# Striatal adenosine A$_{2A}$ receptor neurons control active-period sleep via parvalbumin neurons in external globus pallidus

Xiang-Shan Yuan[1,2,3†], Lu Wang[1,2†], Hui Dong[1,2†], Wei-Min Qu[1,2], Su-Rong Yang[1,2], Yoan Cherasse[4], Michael Lazarus[4], Serge N Schiffmann[5], Alban de Kerchove d'Exaerde[5], Rui-Xi Li[3*], Zhi-Li Huang[1,2*]

[1]Department of Pharmacology, School of Basic Medical Science, Fudan University, Shanghai, China; [2]State Key Laboratory of Medical Neurobiology, Institutes of Brain Science and Collaborative Innovation Center for Brain Science, Fudan University, Shanghai, China; [3]Department of Anatomy, Histology and Embryology, School of Basic Medical Science, Fudan University, Shanghai, China; [4]International Institute for Integrative Sleep Medicine, University of Tsukuba, Tsukuba, Japan; [5]Laboratory of Neurophysiology, ULB Neuroscience Institute, Université Libre de Bruxelles, Brussels, Belgium

*For correspondence:
ruixilee@shmu.edu.cn (R-XL);
huangzl@fudan.edu.cn (Z-LH)

[†]These authors contributed equally to this work

Competing interests: The authors declare that no competing interests exist.

**Abstract** Dysfunction of the striatum is frequently associated with sleep disturbances. However, its role in sleep-wake regulation has been paid little attention even though the striatum densely expresses adenosine A$_{2A}$ receptors (A$_{2A}$Rs), which are essential for adenosine-induced sleep. Here we showed that chemogenetic activation of A$_{2A}$R neurons in specific subregions of the striatum induced a remarkable increase in non-rapid eye movement (NREM) sleep. Anatomical mapping and immunoelectron microscopy revealed that striatal A$_{2A}$R neurons innervated the external globus pallidus (GPe) in a topographically organized manner and preferentially formed inhibitory synapses with GPe parvalbumin (PV) neurons. Moreover, lesions of GPe PV neurons abolished the sleep-promoting effect of striatal A$_{2A}$R neurons. In addition, chemogenetic inhibition of striatal A$_{2A}$R neurons led to a significant decrease of NREM sleep at active period, but not inactive period of mice. These findings reveal a prominent contribution of striatal A$_{2A}$R neuron/GPe PV neuron circuit in sleep control.
DOI: https://doi.org/10.7554/eLife.29055.001

## Introduction

It is widely known that the striatum (caudate putamen), which resides in the forebrain and serves as the primary input nucleus of the basal ganglia, is involved in an array of physiological processes including motor control, habit formation, and goal-directed behaviors (*Durieux et al., 2011*; *Graybiel, 2008*; *Li et al., 2016*). Up to 90% of patients with Parkinson's disease (PD) exhibit severe sleep disturbances, which is one of the most frequent non-motor symptoms (*Arnulf et al., 2008*). Since the substantia nigra pars compacta (SNc), which projects primarily to the striatum, is the major area of degeneration in PD, dysfunction of the striatum may contribute to sleep disturbances in PD patients. However, to date, few studies have examined the role of the striatum in sleep-wake regulation.

The limited reports on the role of the striatum in sleep-wake regulation are controversial. In animal studies, surgical removal of the striatum in cats (*Villablanca, 1972*) and striatal excitotoxic

lesions in rats (*Mena-Segovia et al., 2002*) decrease time spent in sleep. However, electrical lesion of the rat striatum selectively increases rapid eye movement (REM) sleep (*Corsi-Cabrera et al., 1975*), and lesion by ibotenic acid induces an increase in the non-rapid eye movement (NREM) and a decrease in the REM sleep (*Qiu et al., 2010*). In humans, a $H_2^{15}O$ PET study shows that cerebral blood flow (represents activity of the brain) in caudate nucleus, which is equivalent to the rostral striatum of rodents (*Kreitzer, 2009*), increases during REM sleep and decreases during slow-wave sleep (*Braun et al., 1997*). Since lesion and imaging methods exhibit limitations, specific manipulation of neuronal activity with simultaneous electroencephalogram (EEG) recording provides a powerful tool to understand the role of the striatum in sleep-wake cycles.

The striatum contains GABAergic medium spiny neurons (MSNs, 95%) and interneurons (5%) (*Kreitzer, 2009*). MSNs are divided into two projection neuron classes. One class consists of striato-pallidal neurons that express the adenosine $A_{2A}$ receptors ($A_{2A}$Rs) and dopamine $D_2$ receptors ($D_2$Rs), and project to the external globus pallidus (GPe). The other class consists of striatonigral neurons that express adenosine $A_1$ receptors ($A_1$Rs) and dopamine $D_1$ receptors ($D_1$Rs), and project primarily to the substantia nigra pars reticulata (SNr) and the internal globus pallidus (GPi) (*Kreitzer, 2009*). Both $A_1$Rs and $A_{2A}$Rs have been reported to regulate sleep (*Basheer et al., 2004*; *Thakkar et al., 2003*; *Urade et al., 2003*). Among them, $A_1$Rs contribute to sleep induction in a region-dependent manner, whereas $A_{2A}$Rs play a predominant role in sleep induction (*Huang et al., 2005*; *Lazarus et al., 2012*; *Wang et al., 2017*). Moreover, it has been reported that the expression level of $A_{2A}$Rs is altered in the striatum of PD patients (*Mishina et al., 2011*; *Ramlackhansingh et al., 2011*), which may change the activity of striatal $A_{2A}$R neurons (*Gerfen et al., 1990*; *Mitchell et al., 1989*), thus contributing to sleep disturbances in PD patients. However, it is unknown about the role and circuits of striatal $A_{2A}$R neurons in regulation of sleep-wake behavior.

To address these questions, we employed a chemogenetic technique known as designer receptor exclusively activated by designer drugs (DREADD) (*Alexander et al., 2009*), which specifically and non-invasively manipulates neuronal activity based on the principle of Cre/LoxP recombination (*Farrell and Roth, 2013*), and neural tracing, immunoelectron microscopy, as well as optogenetic and electrophysiological methods. We selectively manipulated activity of striatal $A_{2A}$R neurons in *Adora2a*-Cre mice to topographically investigate their contributions to sleep and characterize the functional connectivity between striatal $A_{2A}$R neurons and neurons in the GPe. Then, in *Pvalb*-Cre mice, we studied the role of GPe parvalbumin (PV) neurons, which are downstream targets of $A_{2A}$R neurons, in sleep-wake behavior. In addition, using *Adora2a/Pvalb*-Cre mice expressing DREADD in striatal $A_{2A}$R neurons and selective lesion GPe PV neurons, we examined the neuronal circuit for sleep induced by activation of $A_{2A}$R neurons in the striatum. The results revealed for the first time that $A_{2A}$R neurons in the rostral and central striatum contribute to sleep-wake behavior through the striatal $A_{2A}$R neuron/GPe PV neuron pathway.

## Results

### Chemogenetic activation of $A_{2A}$R neurons in the rostral, centromedial and centrolateral, but not caudal striatum promoted NREM sleep

To test the ability of $A_{2A}$R neurons in the different subregions of the striatum to regulate sleep in mice, we bilaterally injected a Cre-dependent synapsin-driven adeno-associated viral (AAV) vector containing excitatory modified muscarinic Gq-coupled hM3Dq receptors (hSyn-DIO-hM3Dq-mCherry-AAV, *Figure 1—figure supplement 1A*) into the striatum of *Adora2a*-Cre mice. Within the striatum of *Adora2a*-Cre mice, hSyn-DIO-hM3Dq-mCherry-AAV caused robust expression of hM3Dq receptors in $A_{2A}$R neurons (visualized with immunoreactivity for mCherry) (*Figure 1A and E* and *Figure 1—figure supplement 1B and C*). In addition, intraperitoneal (i.p.) injection of the specific hM3Dq ligand, clozapine-N-oxide (CNO, 1 mg/kg), but not vehicle, dramatically drove c-fos expression in hM3Dq-expressing $A_{2A}$R neurons (*Figure 1—figure supplement 1B and C*), which confirmed CNO-induced activation of hM3Dq-expressing $A_{2A}$R neurons in vivo. Moreover, whole-cell current-clamp recordings showed that a brief CNO (5 µM) perfusion induced depolarization and firing of hM3Dq-expressing $A_{2A}$R neurons in the striatum and the effect was reversible following CNO washout (*Figure 1—figure supplement 1D*). In addition, the hSyn-DIO-mCherry-AAV was

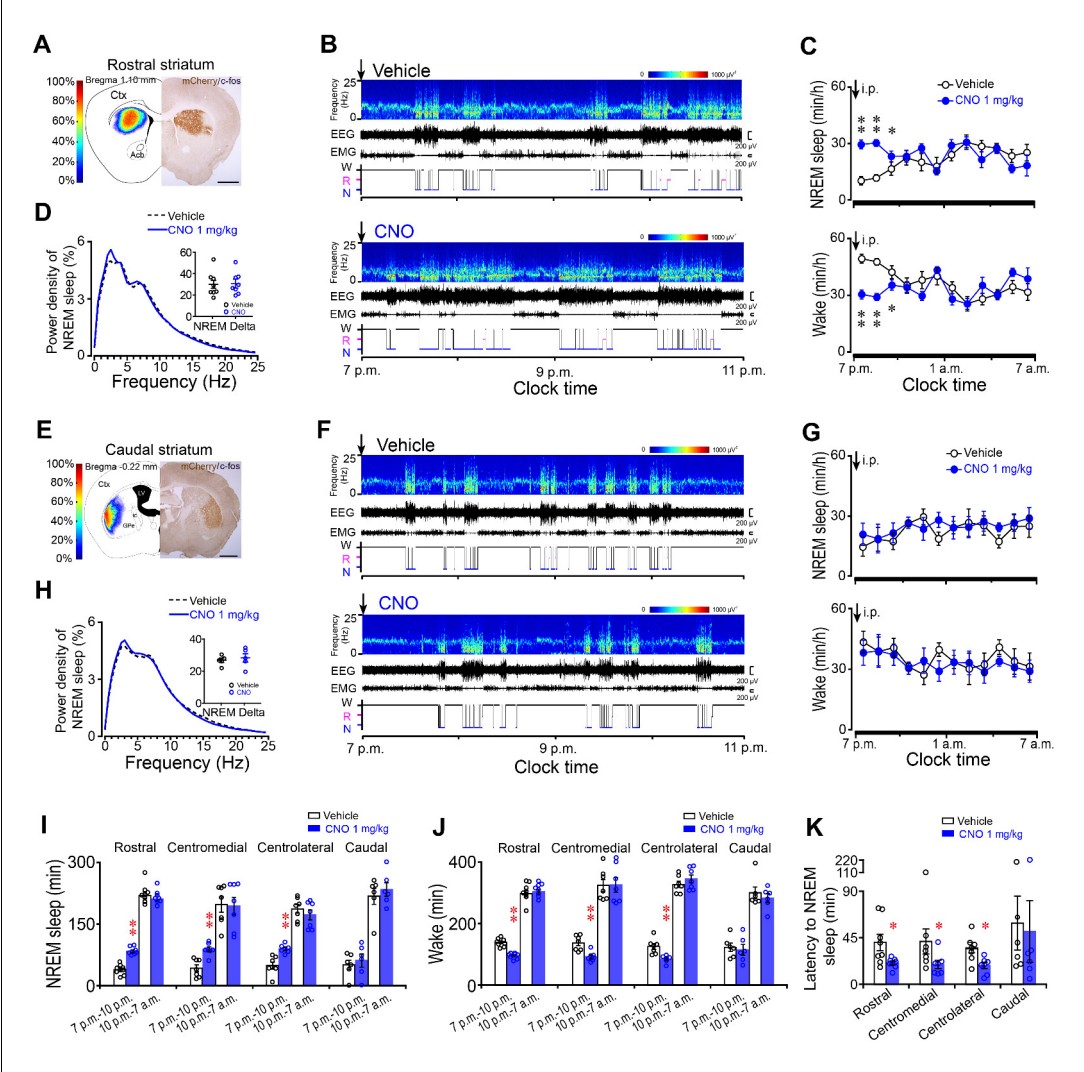

**Figure 1.** Chemogenetic activation of $A_{2A}R$ neurons in the rostral, centromedial and centrolateral, but not caudal, striatum increased NREM sleep during active period. (A, E) Heat map (left) shows the virus-injected area, and immunostaining micrograph (right) represents hM3Dq-expressing neurons (mCherry+) in the rostral (A) or caudal (E) striatum of *Adora2a*-Cre mice. Scale bar, 1 mm. Ctx, cortex; GPe, external globus pallidus; ic, internal capsule; LV, lateral ventricle. (B, F) Typical examples of compressed spectral array (0–25 Hz) EEG, EMG and hypnograms over 4 hr following intraperitoneal (i.p.) administration of vehicle (top panel) or CNO (bottom panel) in a mouse with bilateral hM3Dq receptor expression in $A_{2A}R$ neurons of the rostral (B) or caudal (F) striatum. (C, G) Time course of NREM sleep (top panel) and wakefulness (bottom panel) following vehicle (open black circle) and CNO (closed blue circle) injections (i.p.) in *Adora2a*-Cre mice with hM3Dq-expressing neurons in the rostral (C, NREM: two-way repeated measures ANOVA, n = 8, $F_{1,14}$ = 45.113, p=9.836E-6; paired *t* test, **p=4.999E-5, **p=3.613E-5, *p=0.022. wake: two-way repeated measures ANOVA, n = 8, $F_{1,14}$ = 36.632, p=2.975E-5; paired *t* test, **p=5.919E-5, **p=3.613E-5, *p=0.034.) or caudal (G, NREM: two-way repeated measures ANOVA, n = 6, $F_{1,10}$ = 0.3, p=0.596; wake: two-way repeated measures ANOVA, n = 6, $F_{1,10}$ = 0.16, p=0.697) striatum. (D, H) Relative average EEG power density of NREM sleep during the 3 hr period after CNO or vehicle injections and quantitative changes in power for the delta (0.5–4.0 Hz) frequency bands (insert) following CNO or vehicle in *Adora2a*-Cre mice with hM3Dq-expresssing neurons in the rostral (D, paired *t* test, n = 8, p=0.051) or caudal (H, paired *t* test, n = 6, p=0.356) striatum. (I, J) Total amount of NREM sleep (I) (paired *t* test, Rostral: n = 8, **p=1.518 × 10⁻⁵, p=0.052; Centromedial: n = 7, **p=1.144E-4, p=0.716; Centrolateral: n = 7, **p=0.002, p=0.194; Caudal: n = 6, p=0.169, p=0.366) and wakefulness (J) (paired *t* test, Rostral: n = 8, **p=2.096E-5, p=0.345; Centromedial: n = 7, **p=2.449E-4, p=0.84; Centrolateral: n = 7, **p=0.004, p=0.137; Caudal: n = 6, p=0.276, p=0.362) during the 3 hr post-injection period (7 p.m.–10 p.m.) and the subsequent 9 hr of the active period (10 p.m.–7 a.m.) following vehicle or CNO injections in *Adora2a*-Cre mice expressing hM3Dq receptors in the rostral, centromedial, centrolateral, and caudal striatum. (K) The latency of NREM sleep (paired *t* test, Rostral: n = 8, *p=0.026; Centromedial: n = 7, *p=0.048; Centrolateral: n = 7, *p=0.032; Caudal: n = 6, p=0.313) following vehicle or CNO injections in *Adora2a*-Cre mice expressing hM3Dq receptors in the rostral, centromedial, centrolateral, and caudal striatum. See *Figure 1—source data 1*.
DOI: https://doi.org/10.7554/eLife.29055.002

The following source data and figure supplements are available for figure 1:

**Source data 1.** Sample size (n), mean and SEM are presented for the data in *Figure 1*.

*Figure 1 continued on next page*

*Figure 1 continued*

DOI: https://doi.org/10.7554/eLife.29055.008

**Figure supplement 1.** Expression of hM3Dq receptors in the striatum of *Adora2a*-Cre mice.

DOI: https://doi.org/10.7554/eLife.29055.003

**Figure supplement 2.** Administration of CNO had no effect on the sleep-wake cycle of control mice.

DOI: https://doi.org/10.7554/eLife.29055.004

**Figure supplement 2—source data 1.** Sample size (n), mean and SEM are presented for the data in *Figure 1—figure supplement 2*.

DOI: https://doi.org/10.7554/eLife.29055.009

**Figure supplement 3.** Chemogenetic activation of the $A_{2A}R$ neurons in the rostral, centromedial, centrolateral or caudal striatum did not affect REM sleep.

DOI: https://doi.org/10.7554/eLife.29055.005

**Figure supplement 3—source data 2.** Sample size (n), mean and SEM are presented for the data in *Figure 1—figure supplement 3*.

DOI: https://doi.org/10.7554/eLife.29055.010

**Figure supplement 4.** Chemogenetic activation of $A_{2A}R$ neurons altered the mean duration but not the episode number of NREM sleep.

DOI: https://doi.org/10.7554/eLife.29055.006

**Figure supplement 4—source data 3.** Sample size (n), mean and SEM are presented for the data in *Figure 1—figure supplement 4*.

DOI: https://doi.org/10.7554/eLife.29055.011

**Figure supplement 5.** Chemogenetic activation of $A_{2A}R$ neurons in the centromedial and centrolateral striatum increased NREM sleep during active period.

DOI: https://doi.org/10.7554/eLife.29055.007

**Figure supplement 5—source data 4.** Sample size (n), mean and SEM are presented for the data in *Figure 1—figure supplement 5*.

DOI: https://doi.org/10.7554/eLife.29055.012

injected into the striatum of *Adora2a*-Cre mice, to only express mCherry but not hM3Dq receptors in the $A_{2A}R$ neurons. CNO application had no effect on membrane potential and firing of non-hM3Dq-expressing neurons in the striatum in vitro (data not shown) and sleep-wake profiles of non-hM3Dq-expressing mice in vivo (*Figure 1—figure supplement 2*). The abovementioned results confirmed CNO-mediated activation of striatal neurons only expressing hM3Dq receptors.

Based on the medial-lateral and rostral-caudal axis of the striatum, we injected DREADD-AAV into four subregions (rostral, centromedial, centrolateral and caudal) to investigate their contributions to sleep regulation mediated by striatal $A_{2A}R$ neurons. Following vehicle injection (i.p.) at 7 p.m. (light off), the beginning of the active period when mice usually show high levels of arousal, mice expressing hM3Dq receptors in the rostral striatum (*Figure 1A*) displayed long bouts of wakefulness marked by low EEG slow-wave activity and high electromyogram (EMG) activity (*Figure 1B*). However, CNO (1 mg/kg) injection induced an increase in NREM sleep and remarkably shortened NREM sleep latency (*Figure 1K*), with a decrease in wakefulness for 3 hr (*Figure 1B and C*). The amount of CNO-induced NREM sleep was significantly increased by 2.2-fold during the 3 hr post-injection period as compared with vehicle (*Figure 1I*). Consistent with these results, the amount of wakefulness was significantly decreased by 32% following administration of CNO (*Figure 1B, C and J*), but REM sleep was not significantly changed (*Figure 1—figure supplement 3A and E*). CNO injection induced an increase in the mean duration of NREM sleep with a decrease in wakefulness, and had no effect on the episode numbers of 3 vigilance stages (*Figure 1—figure supplement 4*). Notably, chemogenetic activation of $A_{2A}R$ neurons in the rostral striatum did not change NREM sleep during the subsequent 9 hr of the active period (*Figure 1I*). Along with the increase of NREM sleep amount after CNO injection, however, no change was detected in EEG power density of NREM sleep during the 3 hr post-CNO injection as compared with vehicle (*Figure 1D*), which suggests that the increased sleep was similar to a physiological sleep pattern. Similar to the rostral striatum, chemogenetic activation of $A_{2A}R$ neurons in the centromedial or centrolateral striatum increased NREM sleep by 2.2-fold and 1.8-fold, respectively (*Figure 1I and J* and *Figure 1—figure supplements 3B, C, F, G, 4* and *5*).

Unexpectedly, administration of CNO did not change the total amount, the episode number, the mean duration and the latency of NREM sleep, or EEG power density in mice expressing hM3Dq receptors in the caudal striatum (*Figure 1I*) when compared with vehicle (*Figure 1F-K* and *Figure 1—figure supplement 3D–H and 4D–H*). These results indicate that $A_{2A}R$ neurons in the rostral, centromedial, and centrolateral but not caudal striatum are involved in sleep-wake regulation.

## Topographically organized projections of striatal A$_{2A}$R neurons

To determine whether output pattern varies in different subregions of the striatum, we examined pallidal projections by injecting CMV-lox-stop-hrGFP-AAV (*Figure 2A*) into different subregions of the striatum in *Adora2a*-Cre mice. This virus caused robust expression of humanized *Renilla* green fluorescent protein (hrGFP) in the cytosol of A$_{2A}$R neurons (*Zhang et al., 2013*). Using immunofluorescence, we found that hrGFP-expressing neurons displayed typical morphology of MSNs (*Figure 2B*) in the striatum, and positive signals for A$_{2A}$R immunoreactivity (*Figure 2C–F*). Furthermore, we found 3 types of axonal arrangement based on different virus injection sites in the striatum. In sagittal sections, axons of A$_{2A}$R neurons in the rostral striatum were found in the rostral GPe with a discoidal field paralleling the strio-pallidal border (*Figure 2G*). Axons from the central striatum were distributed not only in the rostral but also the caudal GPe, forming similar discoidal areas paralleling the strio-pallidal border (*Figure 2G*). In contrast, axons from the caudal striatum were distributed only in the caudal GPe (*Figure 2G*). In addition, it is notable that axons of the lateral striatum projected preferentially to the lateral GPe. These findings indicate that the projections of A$_{2A}$R neurons in different subregions of the striatum are organized topographically in the GPe and suggest that the topographical projections of A$_{2A}$R neurons may contribute to the discrepancies in A$_{2A}$R neuron-mediated sleep.

## Striatopallidal terminals formed more symmetric synapses with PV-positive neurons than PV-negative neurons in the GPe

Neurons within the GPe can be divided into PV-positive and PV-negative neurons (*Dodson et al., 2015*). To examine synapses between axon terminals of A$_{2A}$R neurons and PV-positive neurons expressing PV or PV-negative neurons not expressing PV in the GPe, we processed mouse GPe samples expressing hrGFP originating from A$_{2A}$R neurons in the striatum for immunoelectron microscopy. In the GPe, hrGFP-IR elements were filled by floccular diaminobenzidine (DAB) reaction products and represented the terminal of striatal A$_{2A}$R neurons. PV-IR ones were filled with punctate reaction products of Vector very intense purple (V-VIP) and represented the GPe PV neurons (*Li et al., 2002*). However, PV-unlabeled dendrites and PV-unlabeled perikarya in the GPe, were not filled with DAB or VIP reaction products and represented the PV-negative neurons in the GPe. We observed that hrGFP-IR terminals established symmetric synapses with dendrites that were labeled or unlabeled for PV (*Figure 2H and J*). Moreover, a small number of perikarya, which were labeled or unlabeled for PV, received symmetric synapses from hrGFP-IR terminals (*Figure 2I and K*). However, hrGFP-labeled terminals formed significantly more synapses with PV-IR profiles than PV-unlabeled profiles (81.2% vs. 18.8%, n = 377 synapses from five mice; *Figure 2L*) in the rostral GPe. In contrast, in the caudal GPe, hrGFP-IR terminals preferentially formed synapses with PV-unlabeled profiles than PV-labeled profiles (61.3% vs. 38.7%, n = 275 synapses from five mice; *Figure 2L*).

These anatomical findings indicate that A$_{2A}$R neurons in the rostral striatum preferentially form symmetric synapses with PV-positive neurons in the rostral GPe, while A$_{2A}$R neurons in the caudal striatum preferentially form synapses with PV-negative neurons in the caudal GPe, indicating a rostral-caudal variation in striatopallidal connections.

## Optogenetic stimulation of the striatopallidal terminals inhibited GPe neurons

To explore the functional nature of striatopallidal connections, we employed an optogenetic-assisted circuit mapping approach (*O'Connor et al., 2015*). Channelrhodopsin-2 (ChR2), a blue light-gated cation channel, was expressed in A$_{2A}$R neurons by injecting hSyn-DIO-ChR2-mCherry-AAV into the striatum of *Adora2a*-Cre mice (*Figure 3A* and *Figure 3—figure supplement 1A*). After 3 weeks, acute coronal brain slices containing the striatum or GPe were prepared for in vitro patch-clamp recording. We first tested responses from somata of ChR2-expressing neurons, which were presumably A$_{2A}$R neurons exhibiting typical morphological and electrophysiological properties of striatal projecting neurons (*Figure 3—figure supplement 1B–H*). Photostimulation of cell bodies of A$_{2A}$R neurons elicited robust photocurrents (*Figure 3—figure supplement 1I*), and trains of brief blue light flashes (1–3 ms) evoked single action potentials at frequencies of 5–30 Hz with high fidelity (*Figure 3—figure supplement 1J–M*).

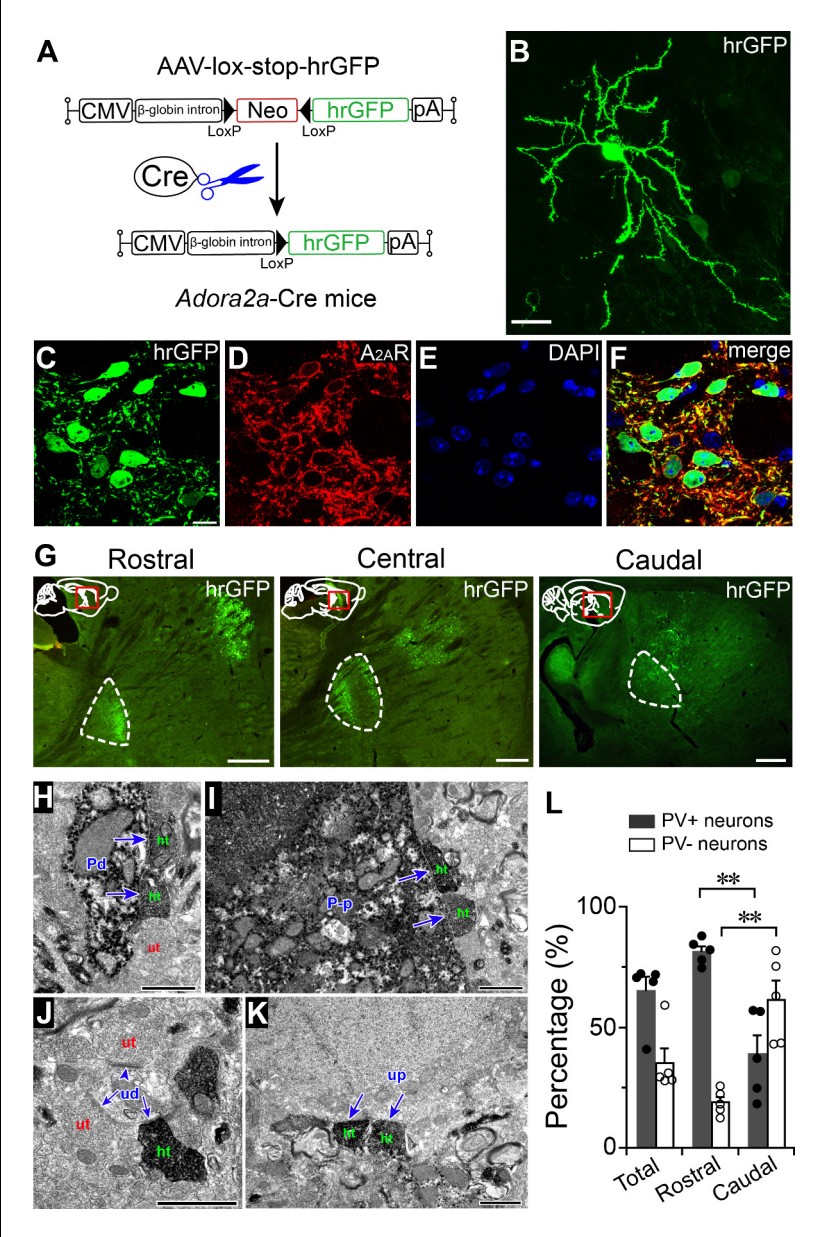

**Figure 2.** Topographical projections of striatal A$_{2A}$R neurons in the GPe. (**A**) Schematic of Cre-dependent hrGFP expression in *Adora2a*-Cre mice. (**B**) Fluorescent micrograph showing a typical A$_{2A}$R neuron that was labeled by injection of hrGFP, in the striatum. Scale bar, 20 μm. (**C–F**) Fluorescent micrographs showing colocalization of hrGFP with A$_{2A}$R immunoreactivity, confirming selective expression of hrGFP in A$_{2A}$R neurons. Scale bar, 10 μm. (**G**) Fluorescent micrographs of sagittal brain sections showing that hrGFP-expressing A$_{2A}$R neurons in the rostral, central, and caudal striatum, send axons to the rostral, rostral plus caudal, and caudal areas of the GPe, respectively. Dotted lines indicate GPe boundaries. Scale bar, 500 μm. (**H–K**) Electron micrographs showed that hrGFP-IR terminals (ht) formed symmetric synapses (arrow) with a PV-IR dendrite (Pd), (**H**), a PV-IR perikaryon (P–p), (**I**), a PV-unlabeled dendrite (ud), (**J**), and a PV-unlabeled perikaryon (up), (**K**) in the GPe. In contrast, hrGFP-unlabeled terminals (ut), (**H** and **J**) formed both symmetric and asymmetric synapses (arrowhead) with PV-IR (Pd), (**H**) and PV-unlabeled dendrites (ud), (**J**). Scale bar, 1 μm (**H–K**). (**L**) Percentages of hrGFP-IR terminals that formed synaptic contacts with PV-positive and PV-negative neurons in the whole, rostral, and caudal GPe (n = 5, paired *t* test, rostral PV+ vs. caudal PV+, **p=0.0025; rostral PV- vs. caudal PV-, **p=0.0034). See *Figure 2—source data 1*.
DOI: https://doi.org/10.7554/eLife.29055.013

The following source data is available for figure 2:

**Source data 1.** Sample size (n), mean and SEM are presented for the data in *Figure 2*.
DOI: https://doi.org/10.7554/eLife.29055.014

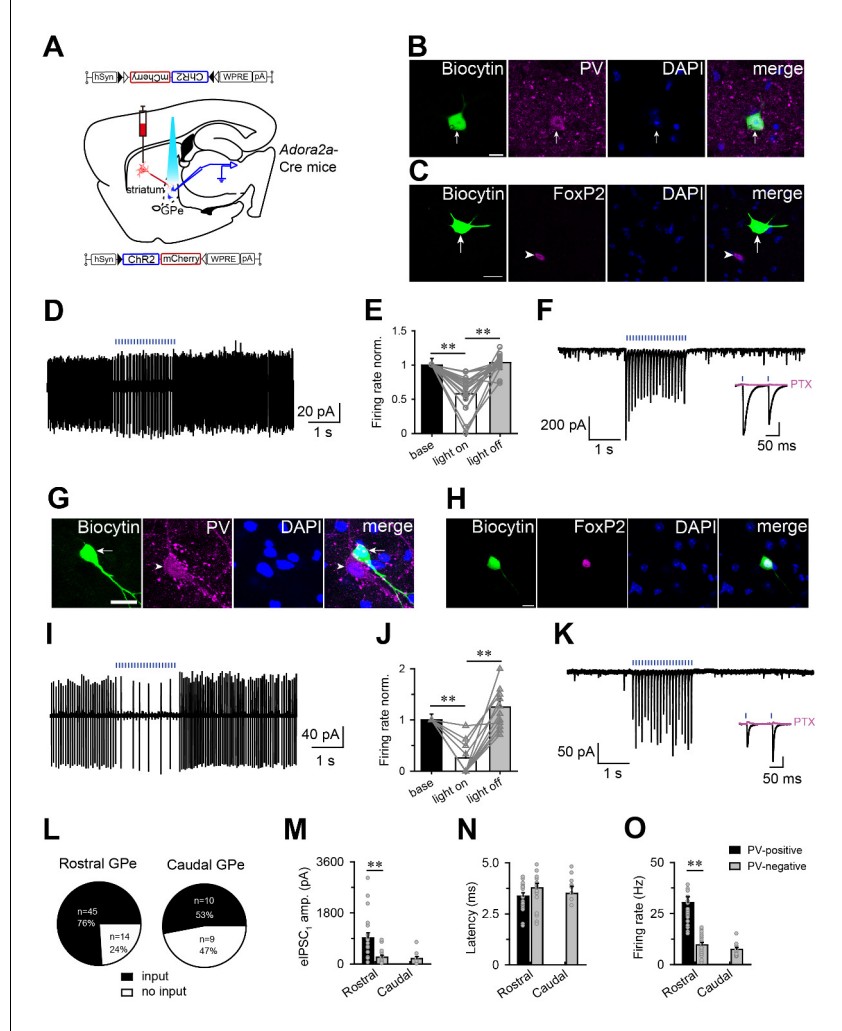

**Figure 3.** Optogenetic activation of striatopallidial terminals suppressed the firing of PV-positive and PV-negative neurons in the GPe. (**A**) Schematic of experiment setup. ChR2-AAV was injected in the striatum of *Adora2a*-Cre mice, and responses were recorded in the GPe. (**B, C**) Fluorescence micrographs showing recorded biocytin-filled GPe neurons that expressed PV (**B**) but not FoxP2 (**C**). Scale bar, 10 μm. (**D, E**) Brief light pulses decreased firing rates of PV-positive neurons in the GPe. Typical cell-attached patch recording (**D**) of a PV-positive neuron. (**E**) Histograms illustrating the normalized firing rate of PV-positive neurons in the GPe before, during and after blue light stimulation in the cell-attached mode (n = 19, paired *t* test, **p=0.001, **p=0.002). (**F**) Photostimulation evoked IPSCs in a GPe PV-positive neuron under voltage-clamp mode. An insert shows that PTX (100 μM; pink line), a GABA$_A$ receptor antagonist, completely abolished the IPSCs. (**G, H**) Fluorescence micrographs showing recorded biocytin-filled GPe neurons that expressed FoxP2 (**H**) but not PV (**G**). Scale bar, 10 μm. (**I, J**) Brief light pulses decreased the firing rate of PV-negative neurons in the GPe. Typical cell-attached patch recording (**I**) from a PV-negative neuron in the GPe. (**J**) Histograms illustrating the normalized firing rate of GPe PV-negative neurons before, during, and after blue light stimulation in the cell-attached patch mode. Brief light pulses decreased the firing rate (n = 12, paired *t* test, **p=2.972E-5, **p=3.465E-4). (**K**) Photostimulation evoked IPSCs in a GPe PV-negative neuron. An insert shows that PTX (100 μM; pink line) completely abolished the IPSCs. (**L**) Number and proportion of recorded neurons in the GPe that did respond to the photostimulation of ChR2-expressing terminals from A$_{2A}$R neurons in the rostral or caudal striatum. (**M–O**) GPe PV-positive neurons showed larger amplitude IPSCs evoked by blue light stimulation (**M**) and higher average firing rate (**O**) with the same latency from light pulse to onset of the evoked IPSC (**N**) compared with GPe PV-negative neurons that received input from A$_{2A}$R neurons in the rostral or caudal striatum. IPSCs: n = 37, independent samples *t* test, **p=0.001; Latency: n = 40, independent samples *t* test, p=0.154; firing: n = 46, independent samples *t* test, **p=0.001. Blue bars in D, F, I, K, represent 1 ms blue light pulses at the frequency of 10 Hz. See ***Figure 3—source data 1***.
DOI: https://doi.org/10.7554/eLife.29055.015

*Figure 3 continued on next page*

*Figure 3 continued*

The following source data and figure supplements are available for figure 3:

**Source data 1.** Sample size (n), mean and SEM are presented for the data in *Figure 3*.
DOI: https://doi.org/10.7554/eLife.29055.019
**Figure supplement 1.** Optogenetic stimulation of A$_{2A}$R neurons in the striatum in vitro.
DOI: https://doi.org/10.7554/eLife.29055.016
**Figure supplement 2.** Optogenetic stimulation of the striatopallidal terminals promoted NREM sleep.
DOI: https://doi.org/10.7554/eLife.29055.017
**Figure supplement 2—source data 1.** Sample size (n), mean and SEM are presented for the data in *Figure 3—figure supplement 2*.
DOI: https://doi.org/10.7554/eLife.29055.018

Next, cells in the GPe were randomly patch-clamped while blue light flashes (1 ms) at 10 Hz were used to stimulate axon terminals of A$_{2A}$R neurons. Previous studies have demonstrated two non-overlapping cell classes in the GPe, one expressing PV, and the other expressing forkhead box protein P2 (FoxP2), a transcription factor (*Abdi et al., 2015*; *Dodson et al., 2015*; *Hernández et al., 2015*; *Mallet et al., 2012*). Thus, to identify the cell type of recorded GPe neurons, we added biocytin to the pipette solution, and performed immunostaining using PV as marker for PV-positive neurons, and FoxP2 as marker for PV-negative neurons, after recording. We found that light-evoked inhibition could be recorded in PV-positive and PV-negative neurons in the GPe. In the cell-attached patch mode, photostimulation decreased the firing rate of most PV-positive neurons, which showed immunoreactive signals for PV but not FoxP2 (*Figure 3B and C*), to 58% of the spontaneous firing rate (from 30.3 ± 2.9 to 17.6 ± 2.6 Hz, n = 19 from 10 mice; *Figure 3D–F*), and the firing rate recovered immediately when photostimulation was terminated. In PV-negative neurons which were FoxP2-IR (*Figure 3G and H*), photostimulation decreased the firing rate to 25% of the spontaneous firing rate (from 9.2 ± 0.9 to 2.3 ± 0.8 Hz, n = 12 from 10 mice; *Figure 3I–K*). Notably, a rebound in the firing rate after photostimulation was observed in 6 of 12 PV-negative neurons (10 mice).

In the whole-cell voltage-clamp mode, flashes of blue light evoked fast inhibitory postsynaptic currents (IPSCs) in both PV-positive and PV-negative neurons (*Figure 3F and K*) with a latency of less than 5 ms (*Figure 3N*), indicating a direct connection between terminals of A$_{2A}$R neurons and PV-positive or PV-negative neurons. In addition, the light-evoked IPSCs were completely abolished by picrotoxin (PTX, 100 μM; *Figure 3F and K*), indicating that these responses were mediated by GABA released from axon terminals of A$_{2A}$R neurons and postsynaptic GABA$_A$ receptors on GPe neurons. Furthermore, light-evoked IPSCs were recorded in 76% of neurons (45 of 59 neurons from 10 mice) in the rostral GPe, and 53% of neurons (10 of 19 neurons from 5 mice) in the caudal GPe (*Figure 3L*). In addition, the amplitude of the first IPSC evoked by blue light flashes at 10 Hz was significantly larger in PV-positive neurons than in PV-negative neurons (860.4 ± 157.3 pA, n = 23 vs. 234.3 ± 69.7 pA, n = 18, from 10 mice) in the rostral GPe (*Figure 3M*). Notably, we did not detect connections between terminals of A$_{2A}$R neurons and PV-positive neuron in the caudal GPe, possibly due to their low numbers in this region. Finally, the spontaneous firing rates of PV-positive and PV-negative neurons recorded in current condition are consistent with previous studies (*Figure 3O*) (*Dodson et al., 2015*). Taken together, these data support anatomical studies indicating that striatal A$_{2A}$R neurons preferentially innervate and inhibit PV neurons in the rostral GPe.

In addition, we examined the effects of photostimulation of ChR2-expressing A$_{2A}$R neuron terminals in the GPe on sleep-wake behavior in freely moving mice. Light stimulation for 1 hr with optical fiber implanted into the GPe, containing ChR2-expressing A$_{2A}$R neuron terminals, remarkably increased NREM sleep by 1.8-fold (*Figure 3—figure supplement 2*), strongly suggesting that striatal A$_{2A}$R neurons promoted sleep by inhibiting neurons, more likely PV neurons, in the GPe.

## Chemogenetic inhibition of PV neurons in the GPe promoted NREM sleep

PV neurons in the GPe serve as an important downstream target for striatal A$_{2A}$R neurons. To test whether PV neurons in the GPe are involved in sleep, we transduced a Cre-recombinase-enabled chemogenetic inhibitory system, a Gi-coupled DREADD, hM4Di receptor, using AAV microinjection into the GPe of *Pvalb*-Cre mice (*Figure 4A*). The hSyn-DIO-hM4Di-mCherry-AAV caused robust and

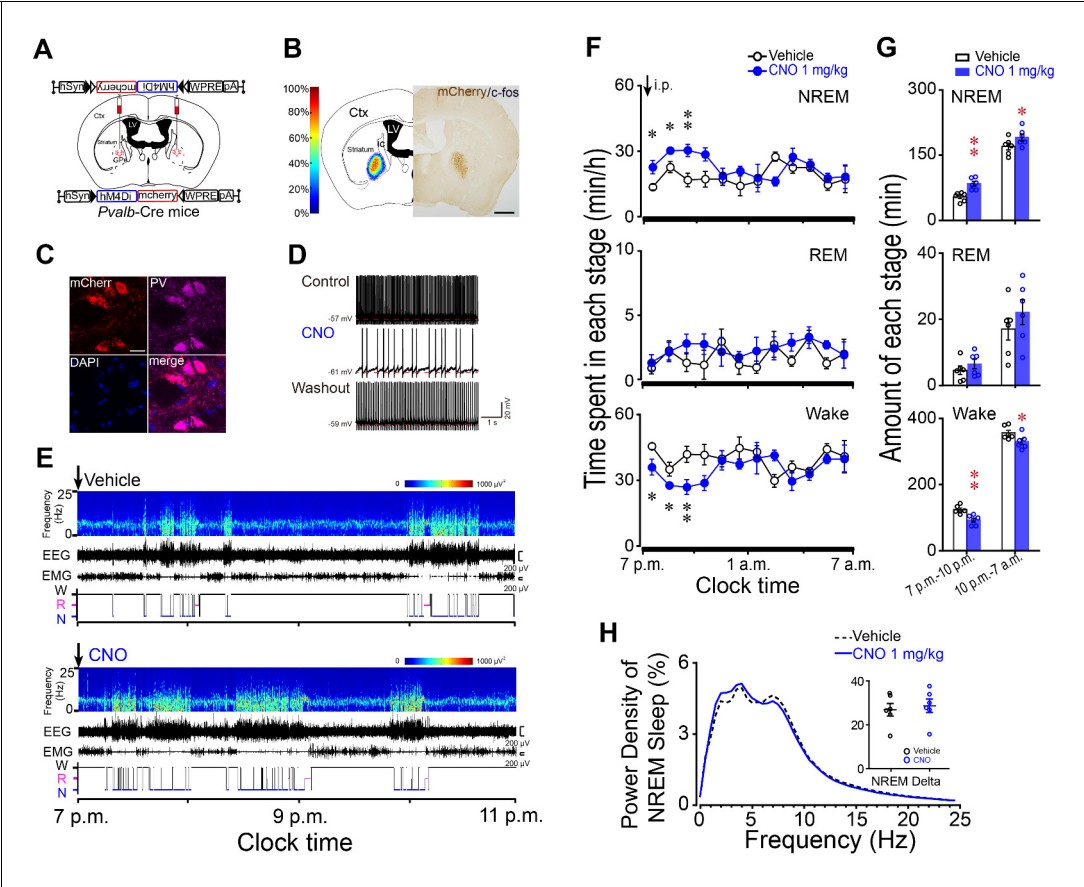

**Figure 4.** Chemogenetic inhibition of PV neurons in the GPe increased NREM sleep during active period. (**A**) Schematic of bilateral virus injection sites in the GPe of *Pvalb*-Cre mice, with details of hSyn-DIO-hM4Di-mCherry-AAV vector injected. (**B**) Heat map (left) shows injected areas, and immunostaining (right) represents hM4Di expression (mCherry+) in the GPe. Scale bar, 1 mm. Ctx, cortex; ic, internal capsule; LV, lateral ventricle. (**C**) Fluorescent micrographs showing colocalization of mCherry with PV immunoreactivity, confirming the selective expression of hM4Di receptors in GPe PV neurons of *Pvalb*-Cre mice. Scale bar, 10 μm. (**D**) Bath applied CNO (5 μM) reduced spontaneous firing rate in hM4Di-expressing PV neurons in the GPe of brain slices. (**E**) Typical examples of compressed spectral array (0–25 Hz) EEG, EMG and hypnograms over 4 hr following administration (i.p.) of vehicle or CNO in a mouse with bilateral hM4Di receptor expression in GPe PV neurons. (**F**) Time course of changes in NREM sleep, REM sleep, and wakefulness following vehicle (open circle) and CNO (closed blue circle) injections during active period in mice expressing hM4Di receptors in PV neurons of the GPe. n = 6, two-way repeated measures ANOVA, paired $t$ test, NREM: $F_{1,10}$ = 18.698, p=0.002 (ANOVA), *p=0.026, *p=0.033, **p=0.001 ($t$-test); REM: $F_{1,10}$ = 1.209, p=0.297 (ANOVA); Wake: $F_{1,10}$ = 14.614, p=0.003 (ANOVA), *p=0.036, *p=0.043, **p=0.002 ($t$-test). (**G**) Total amounts of NREM sleep, REM sleep, and wakefulness during the 3 hr post-injection period (7 p.m.–10 p.m.) and the following 9 hr of the active period (10 p.m.–7 a.m.) following vehicle or CNO injections in mice expressing hM4Di receptors in GPe PV neurons. n = 6, paired $t$ test, NREM: **p=1.972E-4, *p=0.027; REM: p=0.088, p=0.106; Wake: **p=2.508E-4, *p=0.028. (**H**) Relative average EEG power spectrum of NREM sleep and quantitative changes in power for delta (0.5–4.0 Hz) frequency bands (insert) during the 3 hr period after CNO and vehicle injections. n = 6, paired $t$ test, p=0.258. See *Figure 4—source data 1*.

DOI: https://doi.org/10.7554/eLife.29055.020

The following source data is available for figure 4:

**Source data 1.** Sample size (n), mean and SEM are presented for the data in *Figure 4*.

DOI: https://doi.org/10.7554/eLife.29055.021

confined expression of hM4Di receptors in the GPe (*Figure 4B*). Immunofluorescence staining of brain slices with anti-PV antibody revealed that mCherry was exclusively expressed in PV neurons (*Figure 4C*), confirming the requirement for Cre activity to enable expression of hM4Di receptors. Bath application of CNO (5 μM) reduced the spontaneous firing rate of PV neurons expressing hM4Di receptors in the GPe (*Figure 4D*). Systemically, CNO (1 mg/kg, i.p.) injections caused mice to fall asleep with an increased NREM sleep and decreased wakefulness, and this effect was sustained for 3 hr (*Figure 4E–G*). The amount of CNO-induced NREM sleep was significantly increased by 1.6-

fold with a decrease in wake by 26% during the 3 hr post-injection period as compared with vehicle (*Figure 4F and G*). However, REM sleep (*Figure 4F and G*), the latency to the first NREM sleep and EEG power density of NREM sleep (*Figure 4H*) during the 3 hr post-CNO injection was not altered. These findings demonstrate that inhibition of PV neurons in the GPe mimics the effect of activation of striatal $A_{2A}R$ neurons and confirm PV neurons of the GPe as a critical downstream target for striatal $A_{2A}R$ neuron-mediated sleep.

## Lesion of PV neurons in the GPe abolished the increase in NREM sleep caused by activation of striatal $A_{2A}R$ neurons

To test whether $A_{2A}R$ neurons in the striatum promote NREM sleep by innervating PV neurons in the GPe, we crossed *Adora2a*-Cre mice with *Pvalb*-Cre mice to generate *Adora2a/Pvalb*-Cre mice expressing Cre recombinase in $A_{2A}R$ neurons and PV-positive neurons. Using the *Adora2a/Pvalb*-Cre mice, hSyn-DIO-hM3Dq-mCherry-AAV was injected into the rostral and central striatum to express hM3Dq receptors (mCherry+) in $A_{2A}R$ neurons (*Figure 5A–D*). Then, mice, injected with hSyn-DIO-hM3Dq-mCherry-AAV, were microinjected with Flex-taCasp3-TEVp-AAV into the GPe to kill PV-positive neurons (*Figure 5C*) (*Zhang et al., 2016*). Non-lesion mice, injected with hSyn-DIO-hM3Dq-mCherry-AAV, were microinjected with DIO-eGFP-AAV in the GPe (*Figure 5A*). In the GPe, we quantified the number of PV-IR neurons, human neuronal protein HuC/HuD (HuCD)-IR neurons (*Dodson et al., 2015*) which represented total neurons, and FoxP2-IR neurons which represented the PV-negative neurons after three weeks of microinjection of taCasp3 or eGFP viral vector (*Figure 5E–G*). We found that microinjection of taCasp3 vector significantly decreased the number of PV-IR and HuCD-IR neurons compared with the control group (p<0.001; *Figure 5E–G*). In contrast, microinjection of taCasp3 vector did not change the number of FoxP2-IR neurons (p=0.738; *Figure 5E–G*). These data suggested that PV neurons were eliminated in the GPe of the lesion group, but PV-negative neurons were not affected. In freely moving mice, CNO (1 mg/kg) injections caused non-lesion mice that expressed hM3Dq receptors on the striatal $A_{2A}R$ neurons to fall asleep with an increase in NREM sleep lasting for 4 hr (*Figure 5H and J*). The amount of CNO-induced NREM sleep was significantly increased by 2.6-fold, with a decrease in wakefulness by 46% during the 4 hr post-injection period as compared with vehicle (*Figure 5H and J*).

It is very important to note that the PV-lesion mice also expressing hM3Dq receptors in the striatal $A_{2A}R$ neurons did not fall asleep after CNO injection. There was no significant change in NREM sleep and wakefulness following CNO injections in the PV-lesion mice (*Figure 5I and J*). In addition, REM sleep was not changed in both groups following CNO injection (*Figure 5HJ*). Thus, lesion of PV neurons in the GPe abolished the increase in NREM sleep caused by activation of $A_{2A}R$ neurons in the striatum, indicating that the striatal $A_{2A}R$ neurons control sleep behavior via striatal $A_{2A}R$ neurons/GPe PV neurons.

## Chemogenetic inhibition of striatal $A_{2A}R$ neurons induced wakefulness during active period

We demonstrated that chemogenetic activation of $A_{2A}R$ neurons in the rostral, centromedial and centrolateral but not caudal striatum promoted NREM sleep. To examine whether striatal $A_{2A}R$ neurons are necessary for sleep under baseline conditions, we bilaterally injected hSyn-DIO-hM4Di-mCherry-AAV in the rostral and central striatum of *Adora2a*-Cre mice (*Figure 6A, C and D*) to chemogenetically inhibit $A_{2A}R$ neurons. Whole-cell current-clamp recordings of $A_{2A}R$ neurons expressing hM4Di receptors showed decreased firing in response to 350 pA current injection during CNO (5 μM) application, and reversal after washout (*Figure 6B*). Administration of CNO (1 mg/kg) at 7 p.m. (active period) significantly decreased NREM sleep by 70% and REM sleep by 74% in freely moving mice (*Figure 6E–G*), with an increase in wakefulness during the 4 hr post-injection period as compared with vehicle (*Figure 6E–G*). Notably, no sleep rebound was observed in these mice during the subsequent active period (*Figure 6G*). Along with decreases in amount of NREM sleep and REM sleep after CNO injection, no change was detected in the delta power density of NREM sleep and the theta power density of REM sleep during the 4 hr post-CNO injection as compared with vehicle (*Figure 6H and I*). Unexpectedly, administration of CNO at 9 a.m. (inactive period), which is a time of high sleep pressure in mice, did not change time spent in each stage or the power density of NREM sleep, REM sleep, and wakefulness, compared with vehicle (*Figure 6—figure supplement 1*).

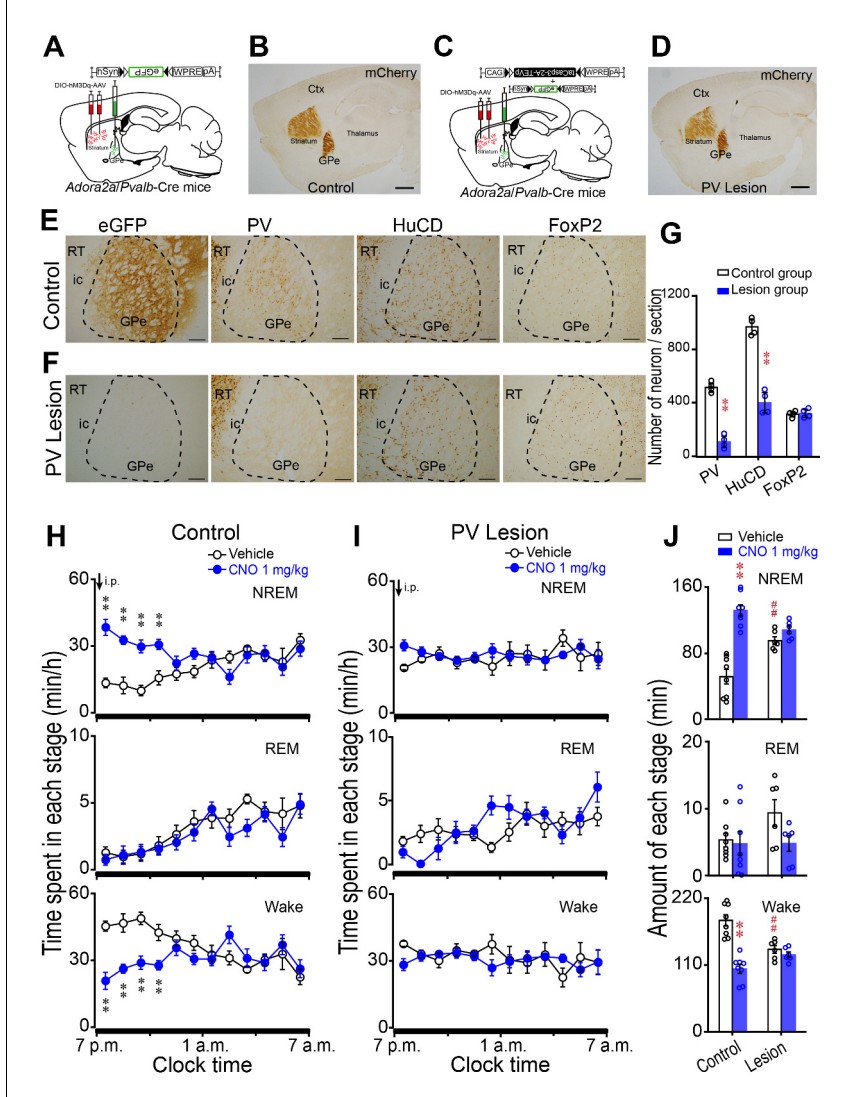

**Figure 5.** Lesion of GPe PV neurons abolished the increase in NREM sleep caused by activation of striatal $A_{2A}R$ neurons. (**A, C**) Schematic of control group (**A**) or lesion group (**C**) by injecting AAV-DIO-hM3Dq in the striatum and AAV-DIO-eGFP in the GPe (**A**) or AAV-DIO-hM3Dq in the striatum and a mixture of AAV-Flex-taCasp3-TEVp and AAV-DIO-eGFP in the GPe (**C**) of *Adora2a/Pvalb*-Cre mice. (**B, D**) Immunostaining micrographs showing hM3Dq expression in $A_{2A}R$ neurons (mCherry+) in the striatum and projections of $A_{2A}R$ neurons in *Adora2a/Pvalb*-Cre mice of the control group (**B**) and lesion group (**D**). Scale bar, 1 mm. Ctx, cortex; GPe, external globus pallidus. (**E, F**) Immunostaining micrographs represent expression of eGFP, PV, HuCD and FoxP2 in the GPe of a *Adora2a/Pvalb*-Cre mouse of control group (**E**) or PV lesion group (**F**). Scale bar, 200 μm. ic, internal capsule; RT, reticular thalamic nucleus. (**G**) The number of PV, HuCD, and FoxP2 neurons within the GPe of a representative coronal section (30 μm thick) in control and lesion mice. n = 4, paired *t* test, PV: **p=0.002; HuCD: **p=0.002; FoxP2: p=0.809. (**H, I**) Time course of changes in NREM sleep, REM sleep, and wakefulness following vehicle (open circle) and CNO (closed blue circle) injections in the control group (**H**, n = 8) and lesion group (**I**, n = 6). Control: n = 8, two-way repeated measures ANOVA, NREM: $F_{1,14}$ = 11.218, p=0.005, paired *t* test, **p=0.002, **p=0.003, **p=1.783E-5, **p=7.75E-6; REM: $F_{1,14}$ = 2.287, p=0.153; Wake: $F_{1,14}$ = 7.511, p=0.016, **p=0.002, **p=0.003, **p=5.551E-5, **p=9.232E-5. Lesion: n = 6, NREM: $F_{1,10}$ = 0.461, p=0.512; REM: $F_{1,10}$ = 0.387, p=0.548; Wake: $F_{1,10}$ = 0.532, p=0.483. (**J**) Total amounts of NREM sleep, REM sleep and wakefulness during the 4 hr post-injection period (7 p.m.–11 p.m.) following vehicle or CNO injections in the control and lesion groups. Vehicle v.s. CNO, control: n = 8, paired *t* test, NREM: **p=5.778E-5; REM: p=0.77; Wake: p=1.21E-5. Lesion: n = 6, paired *t* test, NREM: p=0.171; REM: p=0.078; Wake: p=0.413. Control vehicle v.s. lesion vehicle, independent-samples Student's *t* test, NREM: ## p=0.001; REM: p=0.110; ##Wake: p=3.187E-4. See *Figure 5—source data 1*.

DOI: https://doi.org/10.7554/eLife.29055.022

*Figure 5 continued on next page*

*Figure 5 continued*

The following source data is available for figure 5:

**Source data 1.** Sample size (n), mean and SEM are presented for the data in *Figure 5*.
DOI: https://doi.org/10.7554/eLife.29055.023

---

Therefore, the present results clearly indicate that $A_{2A}R$ neurons in the striatum play a crucial role in maintenance of normal sleep during the active period.

## Discussion

This work constitutes the first investigation of striatal $A_{2A}R$ neuron contributions to regulation of sleep-wake behavior. We showed that activation of $A_{2A}R$ neurons in the rostral and central, but not caudal striatum, promotes NREM sleep during active period. The topographical study revealed that striatal $A_{2A}R$ neurons in the rostral, central, and caudal striatum, send axons to the rostral, rostral plus caudal, and caudal areas of the GPe, respectively. The pathway from striatal $A_{2A}R$ neurons to GPe PV neurons was found to be responsible for sleep control by striatal $A_{2A}R$ neurons. It is worth to note that inhibition of striatal $A_{2A}R$ neurons induces a decrease in sleep during active period, indicating that striatal $A_{2A}R$ neurons are necessary for sustaining sleep during active period.

### Striatal $A_{2A}R$ neuron/GPe PV neuron pathway in sleep regulation

The present study showed that activation of striatal $A_{2A}R$ neurons promotes NREM sleep via the GPe. *Mallet et al. (2012)* identified two major populations of GPe neurons, 'prototypic' and 'arkypallidal' neurons. Most (93%) prototypic neurons expressing PV are fast firing (~50 Hz) compared to arkypallidal neurons expressing FoxP2 (~10 Hz) in the GPe (*Abdi et al., 2015*; *Dodson et al., 2015*). They both fire at a higher rate during active period when compared to slow wave sleep state, thus are wake active (*Abdi et al., 2015*). Using immunoelectron microscopy as well as a combination of optogenetic stimulation and patch-clamp recording, we found that striatal $A_{2A}R$ neurons preferentially form inhibitory synapses with PV-positive neurons (prototypic neurons) in the GPe, suggesting that activation of $A_{2A}R$ neurons in the striatum inhibits PV neurons in the GPe to induce NREM sleep.

In the present study, inhibition of GPe PV neurons mimicked the effect of activation of $A_{2A}R$ neurons in the striatum with an increase in NREM sleep. Most importantly, specific lesions of GPe PV neurons abolished the increase in NREM sleep caused by activation of striatal $A_{2A}R$ neurons. Therefore, $A_{2A}R$ neurons control sleep behavior by innervating PV neurons of the GPe, which indicates the importance of the striatal $A_{2A}R$ neuron/GPe PV neuron pathway in sleep induction. However, whether inhibition of striatal $A_{2A}R$ neurons induced wakefulness also through PV neurons, remains to be investigated in the future.

The mechanism through which PV neurons in the GPe regulate sleep-wake behavior remains unknown. Although the majority of synaptic outputs of GPe PV neurons target the subthalamic nucleus (STN), lesions of the STN have a minimal effect on sleep-wake patterns (*Qiu et al., 2010*), suggesting that PV neurons in the GPe may communicate with other structures in the brain to influence sleep-wake behavior. Recent studies have shown direct GABAergic projections from the GPe to GABAergic interneurons and, to a lesser extent, to pyramidal cells in the cerebral cortex (*Chen et al., 2015*; *Saunders et al., 2015*). Therefore, inhibition of GPe neurons may disinhibit GABAergic interneurons in the cortex to suppress firing of pyramidal cells and promote sleep. Furthermore, it has been reported that GABAergic neurons in the GPe send axons to the thalamic reticular nucleus (TRN) (*Gandia et al., 1993*; *Mastro et al., 2014*) and regulate spiking rates of neurons in the TRN (*Villalobos et al., 2016*). Given that direct activation of GABAergic neurons in the TRN increases the amount of NREM sleep (*Herrera et al., 2016*; *Ni et al., 2016*), we propose that neurons in the GPe, probably PV-positive neurons, may regulate sleep-wake behavior by influencing activity of TRN neurons. Taken together, these results suggest that neural pathways from striatal $A_{2A}R$ neurons to GPe PV neurons and then to cortical interneurons or TRN neurons may be responsible for sleep-wake regulation.

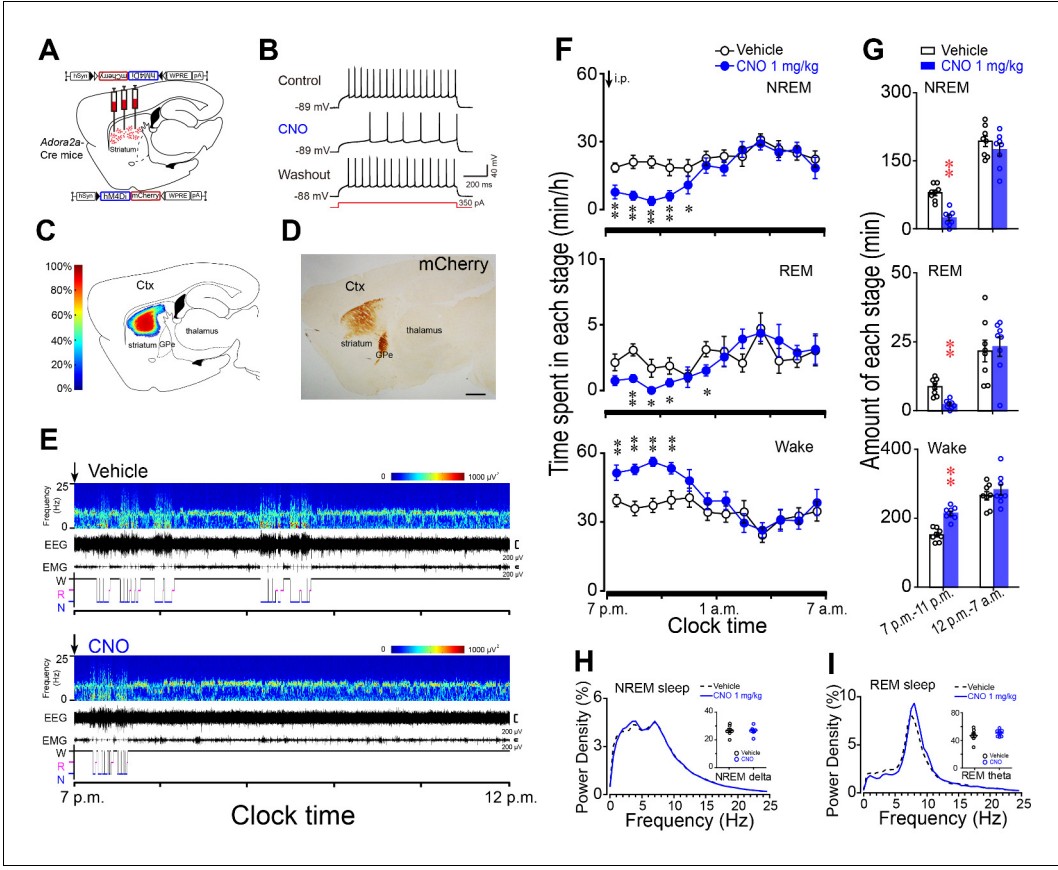

**Figure 6.** Chemogenetic inhibition of striatal A$_{2A}$R neurons reduced NREM sleep during active period. (**A**) Schematic of virus injection into the striatum of *Adora2a*-Cre mice and the expression of hM4Di receptors in A$_{2A}$R neurons. (**B**) Bath applied CNO (5 µM) reduced firing rate in response to 350 pA current injection in hM4Di-expressing A$_{2A}$R neurons of brain slices. (**C, D**) Heat map (**C**) shows the virus-injected areas in the dorsal striatum, and immunostaining (**D**) represents hM4Di-expressing neurons (mCherry+) in striatum. Scale bar, 1 mm. Ctx, cortex; GPe, external globus pallidus. (**E**) Typical examples of compressed spectral array (0–25 Hz) EEG, EMG, and hypnograms over 5 hr following administration (i.p.) of vehicle (top panel) or CNO (bottom panel) in a mouse with bilateral hM4Di expression in striatal A$_{2A}$R neurons. (**F**) Time course of changes in NREM sleep, REM sleep, and wakefulness following vehicle (open circle) and CNO (closed blue circle) injections in *Adora2a*-Cre mice. n = 8, two-way repeated measures ANOVA, and paired *t* test, NREM: F$_{1,14}$ = 19.350, p=6.062E-4, \*\*p=0.003, \*\*p=0.001, \*\*p=2.208E-4, \*\*p=0.003, \*p=0.042; REM: F$_{1,14}$ = 15.221, p=0.002, \*\*p=0.006, \*p=0.045, \*p=0.015; \*p=0.036; Wake: F$_{1,14}$ = 20.302, p=4.937E-4, \*\*p=0.004, \*\*p=0.001, \*\*p=4.055E-4, \*\*p=0.003. (**G**) Total amounts of NREM sleep, REM sleep and wakefulness during the 4 hr post-injection period (7 p.m.–11 p.m.) and the following 8 hr of the active period (11 p.m.–7 a.m.) following vehicle or CNO injections in *Adora2a*-Cre mice expressing hM4Di receptors in the striatum. n = 8, paired *t* test, NREM: \*\*p=2.607E-6; REM: \*\*p=1.653E-4; Wake: \*\*p=1.597E-6. (**H** and **I**) Relative average EEG power density of NREM sleep (**H**) and REM sleep (**I**) during the 4 hr period and quantitative changes in power for delta (0.5–4.0 Hz) frequency bands during NREM sleep (**H**) and theta (6–10 Hz) frequency bands (insert) during REM sleep (**I**) after CNO or vehicle injections in *Adora2a*-Cre mice with hM4Di-expresssing neurons in the striatum. See *Figure 6—source data 1*.

DOI: https://doi.org/10.7554/eLife.29055.024

The following source data and figure supplements are available for figure 6:

**Source data 1.** Sample size (n), mean and SEM are presented for the data in *Figure 6*.
DOI: https://doi.org/10.7554/eLife.29055.027

**Figure supplement 1.** Chemogenetic inhibition of A$_{2A}$R neurons of the striatum did not alter sleep-wake profiles during inactive period.
DOI: https://doi.org/10.7554/eLife.29055.025

**Figure supplement 1—source data 1.** Sample size (n), mean and SEM are presented for the data in *Figure 6— figure supplement 1*.
DOI: https://doi.org/10.7554/eLife.29055.026

## Subregion-specific diversity of striatal A$_{2A}$R neurons in sleep regulation

Subregion-specific diversity of the striatum is not completely understood. Increasing evidence has shown functional heterogeneity along a medial-lateral axis in the striatum in goal-directed action, habit formation, and motor learning (*Durieux et al., 2012*; *Li et al., 2016*; *Rothwell et al., 2015*; *Vicente et al., 2016*); however, little is known about variation along the rostral-caudal axis. A recent study reported that methyl-CpG-binding protein two in the rostral but not caudal striatum, is critical for maintaining local dopamine content and psychomotor control (*Su et al., 2015*). In the present study, activation of A$_{2A}$R neurons in the centromedial or centrolateral striatum induced similar increases in the amount of NREM sleep, suggesting an equal contribution to sleep regulation along the medial-lateral axis in the striatum. However, manipulating A$_{2A}$R neurons in the rostral but not caudal striatum changed the sleep-wake state, strongly suggesting a functional heterogeneity in sleep regulation along the rostral-caudal axis of the striatum.

The mechanisms underlying the functional discrepancy in sleep regulation along the rostral-caudal axis of the striatum remain to be determined. Only one previous study using nonspecific tract tracing in rats showed that the projection of all neurons in the rostral and central striatum is arranged into separate zones of globus pallidus (*Wilson and Phelan, 1982*). On account of limitations of the non-specific method as well as the complexity of striatal subregions and striatal neurons, specific tracing using Cre provides a powerful tool to understand topographical projections of A$_{2A}$R neurons in different subregions, including the rostral, centromedial, centrolateral, and caudal striatum. We revealed the topographical projection of striatal A$_{2A}$R neurons, in which A$_{2A}$R neurons in the rostral, central, and caudal striatum send axons to the rostral, rostral plus caudal, and caudal areas of the GPe, respectively. Combined with the above results from synaptology, we conclude that only A$_{2A}$R neurons in the rostral and central striatum project to the rostral GPe and connect primarily with PV neurons through inhibitory synapses to control sleep behavior. Recently, reports form clinical studies have indicated variation in subregions of the striatum in neurodegenerative diseases, such as PD, and psychiatric disorders, such as bipolar disease (*Altinay et al., 2016*; *Chou et al., 2015*; *Jung et al., 2014*). Our results provide experimental evidence suggesting subregion-specific targets such as the caudate nucleus of human, which is considered equivalent to the rostral striatum of rodents (*Stoffers et al., 2014*; *Su et al., 2015*), for therapeutic intervention of sleep disturbances in clinical cases.

## Necessity of striatal A$_{2A}$R neurons for normal sleep at active period

Homeostatic drive is a major sleep regulating factor. Adenosine, which is released as a neuromodulator in the brain, has been proposed to act as one of the most potent endogenous somnogens to accumulate in the brain during wakefulness and promote physiological sleep through activation of adenosine A$_1$Rs or A$_{2A}$Rs (*Basheer et al., 2004*; *Huang et al., 2005*; *Huang et al., 2014*). Among adenosine receptors, A$_{2A}$Rs play a predominant role in sleep induction, whereas A$_1$Rs contribute to sleep induction in a region-dependent manner (*Huang et al., 2014*). Moreover, A$_{2A}$Rs are present at high concentration in the striatopallidal neurons of the striatum. Evidence has shown that the extracellular adenosine accumulates in the striatum during the active period in rats (*Huston et al., 1996*). Therefore, we predicted that striatopallidal neurons expressing A$_{2A}$Rs may be important in sleep induction of rodents. In the present study, activation of A$_{2A}$R neurons in the striatum induced NREM sleep without any significant change in EEG delta power, suggesting that the induced sleep was similar to physiological sleep.

Interestingly, and somewhat more surprisingly, inhibition of A$_{2A}$R neurons only decreased NREM sleep in active period, but did not alter sleep-wake profiles during the inactive period. During the active period, levels of extracellular adenosine increase in the striatum. Adenosine then acts on excitatory A$_{2A}$Rs expressed on A$_{2A}$R neurons and increases activation of striatopallidal neurons to produce sleep. However, during the inactive period, A$_{2A}$R neurons show low activity because of low levels of extracellular adenosine which results from reduced metabolism of brain tissue and enhanced removal of metabolic products compared with the active period (*Huston et al., 1996*; *Xie et al., 2013*). Thus, chemogenetic inhibition could not further suppress activity of striatal A$_{2A}$R neurons that had been in a state of very low activity during the inactive period. This may explain why the sleep-wake profile was not changed following chemogenetic inhibition of A$_{2A}$R neurons during the inactive period. In addition, we found that inhibition of A$_{2A}$R neurons increased wake without

eliciting a homeostatic sleep response during the active period in mice. Although a homeostatic rebound of sleep following sleep deprivation is a widely accepted phenomenon, our data suggested that increased wake during normal active period is not enough to induce sleep rebound. Together, these findings clearly reveal for the first time that activated $A_{2A}R$ neurons in the striatum are essential for maintaining a certain amount of sleep during the active period and emphasize the importance of striatal $A_{2A}R$ neurons in physiological mechanism of sleep regulation.

### New strategies for treatment of sleep disorders

The present results get access to insight into sleep regulation by striatal $A_{2A}R$ neurons as well as the therapeutic value of striatal $A_{2A}R$ neurons for sleep disturbances related to striatal dysfunction, such as excessive daytime sleepiness (EDS) in PD. The main pathological characteristic of PD is loss of dopaminergic neurons in the SNc, which sends dense dopaminergic innervation to the striatum. In PD patients, EDS has been reported to be a frequent sleep disturbance (*Tholfsen et al., 2015*), but the neuronal mechanisms remain to be elucidated. In humans, *Schulz and Falkenburger (2004)* reported that the degree of dopaminergic terminal loss in the striatum appears to be more pronounced than the magnitude of SNc dopaminergic neuron loss. In a PD animal model, *Gerfen et al. (1990)* demonstrated that mRNA encoding $D_2Rs$ increased in striatopallidal neurons. $D_2Rs$ are localized primarily in dendritic spines of striatopallidal neurons and form heteromers with $A_{2A}Rs$ (*Ferré et al., 2007*). Consistent with $D_2Rs$, levels of $A_{2A}Rs$ also increased in the caudate and putamen of postmortem PD patients (*Villar-Menéndez et al., 2014*). Other studies have reported that $A_{2A}Rs$ in the striatum are over-activated in the PD brain, which may lead to increased activation of striatopallidal neurons ($A_{2A}R$ neurons) in the striatum (*Gerfen et al., 1990*; *Mitchell et al., 1989*), and contribute to EDS. Our findings also suggest that EDS in PD patients may be attributed to over-activation of $A_{2A}Rs$ in the striatum. Therefore, pharmacological antagonism of $A_{2A}Rs$ may be an effective way to treat EDS using a specific $A_{2A}R$ antagonist or a non-specific antagonist such as caffeine (*Chen et al., 2013*; *Huang et al., 2005*; *Rodrigues et al., 2016*).

In summary, our findings demonstrate the importance of striatal $A_{2A}R$ neurons in controlling sleep behavior and reveal subregion diversity underlying this process. Understanding the circuitry through which striatal $A_{2A}R$ neurons contribute to sleep regulation provides insight into the striatal $A_{2A}R$ neuron/GPe PV neuron pathway for sleep regulation and suggests a potential treatment strategy to ameliorate sleep disturbances.

## Materials and methods

### Animals

Pathogen-free adult male (8–10 weeks, 24–28 g) *Adora2a*-Cre mice (*Durieux et al., 2009*) (Tg (Adora2a-cre)2MDkde, RRID:MGI:3852493) on a C57BL/6J background and *Pvalb*-Cre mice (B6; 129P2-Pvalb^tm1(cre)Arbr/J, PV-Cre, RRID:IMSR_JAX:008069) on a 129 background were used (*Do et al., 2016*). *Adora2a*-Cre mice expressing Cre recombinase under the control of adenosine $A_{2A}$ receptor gene (*Adora2a*) promoter were provided by Dr. Serge N. Schiffmann (Université Libre de Bruxelles, Brussels, Belgium) and *Pvalb*-Cre mice expressing Cre recombinase under the control of parvalbumin gene (*Pvalb*) promoter were provided by Dr. Miao He (Fudan University, Shanghai, China). We crossed *Adora2a*-Cre mice with *Pvalb*-Cre mice to generate *Adora2a/Pvalb*-Cre mice expressing Cre in $A_{2A}R$ neurons and PV-positive neurons. The animals were housed in individual cages at constant temperature (22 ± 0.5°C) and relative humidity (60 ± 2%) on an automatically controlled 12:12 light/dark cycle (light on at 7 a.m.) with free access to food and water.

### Surgery

Naïve mice were anesthetized with chloral hydrate (5% in saline, 720 mg/kg) and placed in a stereotaxic apparatus (RWD Life Science, Shenzhen, China). To selectively express Cre-dependent hM3Dq receptors (for sleep-wake monitoring) or hrGFP (for anterograde tract tracing) in $A_{2A}R$ neurons of different subregions in the striatum, hSyn-DIO-hM3Dq-mCherry-AAV, hSyn-DIO-mCherry-AAV or CMV-lox-stop-hrGFP-AAV was delivered bilaterally into the rostral, centromedial, centrolateral, or caudal area of the striatum (coordinates: AP = +1.2 mm, L = ±1.5 mm, DV = −2.4 mm; AP = +0.6 mm, L = ±1.5 mm, DV = −2.4 mm; AP = +0.6 mm, L = ±2.3 mm, DV = −2.4 mm; AP = −0.2 mm, L

= ±2.5 mm, DV = −2.4 mm, respectively, as per the mouse atlas of Paxinos and Keith [*George and Franklin, 2001*]) of *Adora2a*-Cre mice. To selectively express Cre-dependent hM4Di receptors (for sleep-wake monitoring) in $A_{2A}R$ neurons of the rostral and central striatum, hSyn-DIO-hM4Di-mCherry-AAV was delivered bilaterally into the rostral and central striatum (coordinates: AP = +1.2 mm, L = ±1.5 mm, DV = −2.4 mm; AP = +0.7 mm, L = ±1.6 mm, DV = −2.4 mm; AP = +0.3 mm, L = ±2.3 mm, DV = −2.4 mm) of *Adora2a*-Cre mice. Another group of *Adora2a*-Cre mice was microinjected into the striatum with hSyn-DIO-ChR2-mCherry-AAV to selectively express Cre-dependent ChR2. hSyn-DIO-hM4Di-mCherry-AAV was microinjected into the GPe (coordinates: AP = −0.3 mm, L = ±2.5 mm, DV = −3.0 mm) of *Pvalb*-Cre mice for sleep-wake monitoring. hSyn-DIO-hM3Dq-mCherry-AAV was injected into the striatum (coordinates: AP = +1.2 mm, L = ±1.5 mm, DV = −2.4 mm; AP = +0.8 mm, L = ±1.8 mm, DV = −2.4 mm), and Flex-taCasp3-TEVp-AAV or DIO-eGFP-AAV was injected into the GPe of *Adora2a/Pvalb*-Cre mice for sleep-wake monitoring (*Zhang et al., 2016*). Injections of the viral vector (100–200 nL) were performed using a compressed air delivery system as previously described (*Chen et al., 2016*).

Three weeks after injections, mice used for in vivo studies were chronically implanted with EEG and EMG electrodes for polysomnographic recordings under chloral hydrate (5% in saline, 720 mg/kg) anesthesia. The EEG electrode implant consisted of 2 stainless steel screws (1 mm diameter) inserted through the skull into the cortex (anteroposterior, −1.0 mm and left/right, −1.5 mm from bregma). The EMG electrodes consisted of two insulated stainless steel, Teflon-coated wires that were bilaterally placed into both trapezius muscles. All electrodes were attached to a microconnector and fixed to the skull with dental cement (*Qu et al., 2010*). For optogenetic stimulation, the fiber optic cannula (200 µm diameter; Newton Inc., Hangzhou, China) was placed in the GPe: 0.4 mm posterior and 2.2 mm lateral to bregma, 3.3 mm deep from the skull and fixed on the skull using dental cement. The scalp wound was closed with surgical sutures, and each mouse was kept in a warm environment until it resumed normal activity as previously described (*Qu et al., 2010*).

## Sleep-wake monitoring

After the surgical procedures, animals were allowed to recover in individual chambers for at least 7d. Then, each animal was transferred to an insulated sound-proofed recording chamber and connected to an EEG/EMG headstage. The recording cable was connected to a slip-ring device (CFS-22) so the mice could move freely in their cages without tangling the cable. Mice were habituated to the recording cable for 3–4 d before polygraphic recordings. For chemogenetics, EEG/EMG signals mice were recorded over a 24 hr baseline period. This was followed by injection of vehicle (saline, i. p.) at 7:00 p.m. (active period) or 9:00 a.m. (inactive period). Administration of CNO (1 mg/kg in saline, LKT lab, Saint Paul, USA) was performed 24 hr after vehicle injection, and EEG/EMG signals were recorded over the next 24 hr.

## Sleep scoring and analysis

EEG and EMG signals were amplified and filtered (EEG, 0.5–25 Hz; EMG, 20–200 Hz), digitized at a sampling rate of 128 Hz, and recorded using SleepSign for Animal (Kissei Comtec) (*Huang et al., 2005*). When completed, SleepSign was used to automatically scored polygraphic recordings off-line (10 s epochs for chemogenetics and 4 s epochs for optogenetics) into the 3 stages of wakefulness, REM sleep, and NREM sleep with the assistance of spectral analysis using fast Fourier transform (*Huang et al., 2005*; *Qu et al., 2010*). Briefly, NREM sleep was identified by a preponderance of high-amplitude, low frequency (<4 Hz) EEG activity and relatively low and unchanging EMG activity, whereas wakefulness was characterized by a preponderance of low-amplitude, fast EEG activity and highly variable muscle tone on EMG. REM sleep was identified by very low EMG activity and a low amplitude monotonous EEG containing a predominance of theta range (6–9 Hz) EEG activity. As a final step, defined sleep-wake stages were examined visually, and corrected, if necessary. The latency to NREM sleep are defined as the time between the end of the injection and the onset of the first NREM sleep episode lasting >20 s (*Ulbrich et al., 2013*). Scoring was done before histological examination and so the scorers were unaware of the extent of the receptor expression. The amount of time spent in wake, NREM, and REM sleep were determined from the scored data.

EEG power spectra were computed for consecutive 10 s or 4 s epochs within the frequency range of 0.5–25 Hz (0.5 Hz bins for 10 s and 0.25 Hz bins for 4 s), using a fast Fourier transform routine. To

analyze EEG frequency bands, relative power bins were summed as follows: delta = 0.5–4 Hz, theta = 6–10 Hz, alpha = 12–14 Hz and beta = 15–25 Hz (*Chen et al., 2016*). The EEG power spectrum data are expressed as relative values to total power of NREM or REM sleep, and wakefulness. Two mice power spectrum data were removed, as there was no NREM and REM sleep following vehicle or CNO injection over 3 hr or 4 hr (*Figure 1H*; *Figure 6H and I*).

## Optogenetic stimulation in vivo

The optical fiber cannula was attached to a rotating joint (FRJ_FC-FC, Doric Lenses, Canada) to relieve torque. The joint was connected via a fiber to a 473 nm blue laser diode (Newton Inc., Hangzhou, China). Light pulses were generated through a stimulator (SEM-7103 Nihon Kohden, Japan) and output via an isolator (ss-102J, Nihon Kohden). For 1 hr photostimulation, we used programmed light pulse trains (5 ms pulses at 20 Hz for 50 s with 40 s intervals for 1 hr). Light stimulation was conducted from 9 p.m. to 10 p.m. EEG/EMG recorded during the same period on the previous day served as baseline. Light intensity was tested by a power meter (PM10, Coherent) before each experiment and calibrated to emit 20–30 mW/mm$^2$ from the tip of the optical fiber cannula.

## Spectral analysis-compressed spectral array

EEG power spectra of wake as well as NREM and REM epochs were analyzed offline using fast Fourier transformation (256 points, Hanning window, 0–24.5 Hz with 0.5 Hz resolution using SleepSign). Then the EEG power spectra data were converted into a dataset in 10 s epochs disregarded stages. Spectral analysis-compressed spectral array were created for ranges of 0–25 Hz and were 4 hr or 5 hr in length. An amplitude bar graph was simultaneously created as the average of 10 s epochs. Bit maps were exported in TIFF format for generation of figures by MATLAB (The MathWorks Inc., New York, USA) (*Litvak et al., 2011*).

## Immunohistochemistry

Animals received CNO (1 mg/kg, 7 p.m.) and were killed 2 hr later by deep anesthesia with chloral hydrate (10%, 360 mg/kg). This was followed by transcardial perfusion with 10 mL saline, and then 100 mL of 4% paraformaldehyde in 0.1 M phosphate buffer (PB). The brains were removed, postfixed for 4–6 hr at 4°C, and then cryoprotected in 10%, 20%, and 30% sucrose in 0.1 M PB at 4°C until they sank. Tissues were embedded in OCT compound, and stored at −70°C before use (*Li et al., 2001*). The brains were coronally cut at a thickness of 30 μm on a cryostat (Leica 1950) in four series and were collected in 0.01 M phosphate-buffered saline (PBS, pH 7.4). The floating sections were double immunostained according to the following series of incubation steps (*Anaclet et al., 2014*): (1) 1:10000 diluted rabbit polyclonal anti-c-fos antibody (catalog number: ABE457, Millipore, RRID:AB_2631318) containing 3% normal donkey serum (v/v) and 0.25% Triton X-100 (v/v) for overnight at 4°C; (2) donkey anti-rabbit biotinylated IgG (1:1000, Jackson ImmunoResearch) for 2 hr at room temperature (RT); (3) ABC complex (Vectorlab, USA) for 2 hr at RT; (4) 3, 3′ diaminobenzidine (DAB, 0.2 mg/ml) and hydrogen peroxide (0.005%) in 0.05 M Tris buffer containing NiCl according to the manufacturer's instructions (Vector Lab); (5) 1:8000 diluted rabbit polyclonal anti-mCherry antibody (catalog number: 632496, Clonetech, RRID:AB_10013483) containing 3% normal donkey serum (v/v) and 0.25% Triton X-100 (v/v) for overnight at 4°C; (6) donkey anti-rabbit biotinylated IgG (1:1000, Jackson ImmunoResearch) for 2 hr at RT; (7) ABC complex for 2 hr at RT; 8) DAB (0.2 mg/ml) and hydrogen peroxide (0.005%) in 0.05 M Tris buffer without NiCl. Sections were then mounted onto slides, dehydrated, and coverslipped.

Next, injection sites were viewed under high and low magnification to discern the regions of mCherry immunoreactivity. Under low magnification, histological sections were sampled and rotated to match the mouse atlas, and outlines containing the injection regions were then drawn and merged using Matlab software. A color map showed regions of overlap, and the number of overlapping regions was displayed in decreasing order with red as the maximum followed by yellow, gray, and blue.

The floating sections from GPe lesion and control mice were incubated overnight at 4°C in PBS containing 3–5% normal donkey serum (v/v), 0.25% Triton X-100 (v/v) and primary antibodies that were as follows: rabbit anti-eGFP (1:2000, catalog number: A-11122, Life Technologies, RRID:AB_221569), goat anti-PV (1:3000, catalog number: PVG213, Swant, RRID:AB_2650496), mouse anti-

human neuronal protein HuC/HuD (HuCD) (1:500, catalog number: A-21271, Life Technologies, RRID:AB_221448), and goat anti-forkhead box protein P2 (FoxP2) (1:500, catalog number: sc-21069, Santa Cruz, RRID:AB_2107124). After several washes in PBS, the sections were incubated with the appropriate secondary antibody (biotinylated IgG) and ABC complex, followed by DAB (0.2 mg/mL) staining. Sections were then mounted onto slides, dehydrated and coverslipped. Image from the GPe were acquired at 10 × on an Olympus IX71 microscope (Olympus, Japan). The numbers of PV, HuCD and FoxP2 immunoreactive neurons in the GPe (AP = $-0.2 \sim -0.5$ mm from bregma) were counted offline using Image-Pro Plus. The average numbers of PV, HuCD, and FoxP2 immunoreactive neurons in each hemisphere were then analyzed statistically as described below.

To map topographical projections of striatal $A_{2A}R$ neurons, Adora2a-Cre mice, injected with CMV-lox-stop-hrGFP-AAV, were anesthetized with chloral hydrate (10%, 360 mg/kg). Brain tissues were prepared as described above. Sagittal or coronal sections of the striatum and GPe were incubated overnight at 4°C in PBS containing 3–5% normal donkey serum (v/v), 0.25% Triton X-100 (v/v), and a mixture of the following primary antibodies: rabbit anti-hrGFP (1:4000, catalog number: 240142, Agilent, RRID:AB_10644103), goat anti-$A_{2A}R$ (1:1000, catalog number: sc-7504, Santa Cruz, RRID:AB_2273960), and rabbit anti-mCherry (1:8000, Clontech), goat anti-PV (1:3000, Swant). After several washes in PBS, the sections were incubated with Alexa Fluor-conjugated IgG antibody (Invitrogen) at RT for 2 hr. The sections were then incubated in PBS containing DAPI (1:3000, Sigma-Aldrich) for 10 min. Finally, sections were washed in PBS and coverslipped with Fluoromout-G™ (Southern Biotech). Control staining with 0.01 M PBS instead of the primary antisera indicated a complete lack of non-specific staining regardless of the detection methods used. Fluorescence images were collected using a Leica confocal system or Olympus IX71 microscope.

## Electron microscopy

Under deep anesthesia with chloral hydrate (10%, 360 mg/kg), Adora2a-Cre mice injected with AAV-CMV-lox-stop-hrGFP, were briefly perfused transcardially with 6–8 mL saline followed by 100 mL ice cold fixative containing 4% paraformaldehyde, 0.5% glutaraldehyde, and 15% saturated picric acid in 0.1 M PB (pH 7.4). Immediately after perfusion-fixation, brains were removed and postfixed in 4% paraformaldehyde for an additional 2 hr at 4°C.

Brain samples containing the striatum and GPe were cut into coronal sections (40 μm thick) with a vibratome (VT1000S, Leica) and collected in 0.05 M PB for hrGFP and PV double staining. The basic immunohistochemical protocols were the same as those described above. Briefly, the sections were placed in 0.05 M PB (pH 7.4) containing 25% (w/v) sucrose and 10% (v/v) glycerol for 1 hr and then freeze-thawed with liquid nitrogen to enhance antibody penetration (*Li et al., 2002*; *Li et al., 2001*). The sections were then incubated in rabbit anti-hrGFP antibody (1:3000, Agilent) in 0.05 M PB containing 5% normal donkey serum for 30–36 hr at 4°C. Next, the sections were incubated with donkey anti-rabbit biotinylated IgG (1:1000, Jackson ImmunoResearch) for 3 hr at RT, followed by incubation in ABC complex for 3 hr at RT. After hrGFP immunoractivity was visualized with DAB (Vector Laboratories) following the same procedure used for light microscopy, sections were incubated in goat anti-PV antibody (1:2000, Swant) containing 5% normal donkey serum for 30–36 hr at 4°C. The sections were then incubated in donkey anti-goat biotinylated IgG (1:1000, Jackson ImmunoResearch), followed by ABC complex for 3 hr each at RT. PV immunoreactivity was visualized using a chromogen of the Vector VIP substrate kit (Vector Laboratories) (*Li et al., 2002*; *Smiley and Mesulam, 1999*).

The double-labeled sections were osmicated with 2% OsO4, dehydrated in a graded series of ethanol, and embedded flat in Epon 12 (Ted Pella, Redding, USA) using embedding capsules (TAAB, Berks, UK). The sections embedded in Epon were observed under a light microscope and areas of the GPe were sampled. Ultrathin sections were cut at a thickness of 70 nm using an ultramicrotome, stained with uranyl acetate and lead citrate, and then examined in a CM-120 transmission electron microscope (Philips, Netherlands).

## In vitro electrophysiology

Three weeks after injection of hSyn-DIO-ChR2-mCherry-AAV in Adora2a-Cre mice, slices from the striatum or GPe were prepared for electrophysiological recordings. Mice were anesthetized with chloral hydrate and transcardially perfused with ice-cold cutting artificial cerebrospinal fluid (ACSF)

containing (in mM): 213 sucrose, 2.5 KCl, 1.25 $NaH_2PO_4$, 26 $NaHCO_3$, 10 glucose, 2 Na-Pyruvate, 0.4 ascorbic acid, 3 $MgSO_4$, 0.1 $CaCl_2$ (pH 7.3 when carbogenated with 95% $O_2$ and 5% $CO_2$). The brains were quickly removed and cut in coronal slices (300 μm thick) in ice-cold cutting ACSF using a vibrating microtome (VT 1200S, Leica). Slices containing the striatum or GPe were transferred to recording ACSF containing (in mM): 126 NaCl, 2.5 KCl, 1.25 $NaH_2PO_4$, 1 $MgSO_4$, 2 $CaCl_2$, 26 $NaHCO_3$, and 25 Glucose. Slices were incubated at 32°C for 30 min and subsequently maintained at RT for 30 min before recording.

During recording, slices were submerged in a low-volume (~170 μL) recording chamber and continuously superfused at 2 mL/min with warm (30–32°C) ACSF containing 25 μM d-(-)−2-amino-5-phosphonopentanoic acid (d-APV) and 5 μM NBQX to block NMDA and AMPA receptors, respectively. Recordings were guided using a combination of fluorescence and infrared differential interference contrast (IR-DIC) video microscopy using a fixed stage upright microscope (BX51WI, Olympus) equipped with a water immersion lens (40×/0.8 W) and an IR-sensitive CCD camera (IR1000, DAGE MTI). Recording electrodes (3–5 MΩ) were filled with an internal solution consisting of (in mM): 105 potassium gluconate, 30 KCl, 4 ATP-Mg, 10 phosphocreatine, 0.3 EGTA, 0.3 GTP-Na, 10 HEPES (pH 7.3, 295–310 mOsm). The internal solution also contained 0.2% biocytin. Random recordings were obtained from neurons in the striatum expressing mCherry and from neurons in the region of the GPe innervated by $A_{2A}R$ neurons expressing mCherry. Recordings were conducted in the cell-attached or whole-cell configuration using a Multiclamp 700B amplifier (Molecular Devices), a Digidata 1440A interface and Clampex 10.3 software (Molecular Devices). Optical stimulation was delivered to slices via an optical fiber (200 μm core, Thorlabs, Newton, USA) coupled to 470 nm diode-pumped solid-state continuous-wave laser system (OEM Laser Systems Salt Lake City, USA). Stimulation consisted of either a single 1 ms pulse or trains of 1 ms pulses delivered at 10 Hz. Output of the laser was <2 mW. Light-evoked responses and the effects of a $GABA_A$ receptor antagonist (picrotoxin, PTX, 100 μM) on these responses were recorded at a holding potential of −70 mV. All drugs were dissolved in ACSF. Series resistance (Rs) compensation was not used. Therefore, cells with Rs changes over 25% were discarded.

Thirty-two individual GPe neurons were intracellularly labeled with biocytin. Following whole-cell recordings, slices containing cells injected with biocytin were stored in a 4% paraformaldehyde solution overnight at 4°C and then rinsed with PBS. To test for expression of PV or FoxP2, brain slices were incubated in goat anti-PV antibody (1:2000, Swant) or goat anti-FoxP2 (1:500, Santa Cruz) containing 3% normal donkey serum (v/v), 0.5% Triton X-100 (v/v) for 48 hr at 4°C. This was followed by incubation with Alexa Fluor 647-conjugated donkey anti-goat (1:1000; Invitrogen) and Alexa Fluor 488 streptavidin (1:1000, Invitrogen) for 12 hr at RT. The slices were then incubated in PBS containing DAPI (1:3000) for 20 min. Finally, sections were washed in PBS and coverslipped with Fluorom-out-G™.

## Statistics

All data subjected to statistical analysis in SPSS 19.0. The hourly amounts of the amounts of time spent in each stage of the sleep-wake profiles after vehicle or CNO injection were compared using two-way repeated-measures ANOVA and two-tailed paired Student's *t* tests. Two-tailed paired Student's *t* tests were used to compare (1) histograms of amounts of sleep and wakefulness and the latency to the first NREM sleep after vehicle or CNO injection, (2) the hourly amount of sleep and wakefulness after photostimualtion, (3) the percentage of hrGFP-immunoreactive terminals that formed synapses with PV-positive or PV-negative structures in the rostral or caudal GPe, (4) firing rates of PV-positive and PV-negative neurons in the GPe after baseline, light on, or light off, (5) the number of PV, HuCD, or FoxP2 neurons in the GPe of the control group or lesion group, (6) the effect of vehicle or CNO injection on EEG power density bands (delta, 0.5–4 Hz; theta, 6–10 Hz; alpha, 12–14 Hz; beta, 15–25 Hz). eIPSC amplitude, action potential latency, or firing rate of PV-positive and PV-negative neurons in the GPe after light on, and histograms of amounts of sleep and wakefulness in the control and lesion group were compared using Levene's test followed by independent-samples Student's *t* tests. Data are expressed as mean ± SEM. Statistical significance was considered with $p < 0.05$.

## Acknowledgements

We thank Dr. Miao He (Institutes of Brain Science, Fudan University) for kindly supplying us with *Pvalb*-Cre mice. We thank Dr. Jun Lu of Harvard Medical School for helpful discussions. We also thank Dr. Jin-Lian Li of the Fourth Military Medical University for help with the immunoelectron microscopy.

## Additional information

### Funding

| Funder | Grant reference number | Author |
|---|---|---|
| National Natural Science Foundation of China | 81420108015 | Zhi-Li Huang |
| National Basic Research Program of China | 2015CB856401 | Zhi-Li Huang |
| National Natural Science Foundation of China | 31571103 | Lu Wang |
| National Natural Science Foundation of China | 31671099 | Wei-Min Qu |
| National Natural Science Foundation of China | 81271466 | Rui-Xi Li |
| National Natural Science Foundation of China | 31471064 | Zhi-Li Huang |
| National Natural Science Foundation of China | 31530035 | Zhi-Li Huang |
| National Natural Science Foundation of China | 81571296 | Su-Rong Yang |

The funders had no role in study design, data collection and interpretation, or the decision to submit the work for publication.

### Author contributions

Xiang-Shan Yuan, Conceptualization, Formal analysis, Investigation, Visualization, Methodology, Writing—original draft, Writing—review and editing; Lu Wang, Conceptualization, Formal analysis, Funding acquisition, Investigation, Visualization, Methodology, Writing—original draft, Writing—review and editing; Hui Dong, Conceptualization, Software, Formal analysis, Investigation, Methodology, Writing—original draft, Writing—review and editing; Wei-Min Qu, Resources, Supervision, Funding acquisition, Project administration, Writing—review and editing; Su-Rong Yang, Resources, Software, Funding acquisition, Writing—review and editing; Yoan Cherasse, Michael Lazarus, Serge N Schiffmann, Alban de Kerchove d'Exaerde, Resources, Writing—review and editing; Rui-Xi Li, Zhi-Li Huang, Conceptualization, Resources, Supervision, Funding acquisition, Project administration, Writing—review and editing

### Author ORCIDs

Michael Lazarus https://orcid.org/0000-0003-3863-4474
Zhi-Li Huang https://orcid.org/0000-0001-9359-1150

### Ethics

Animal experimentation: All animal studies were performed in accordance with protocols approved by the Committee on the Ethics of Animal Experiments of Fudan University Shanghai Medical College (permit number: 20110307-049). Every effort was made to minimize the number of animals used and any pain and discomfort experienced by the subjects.

Decision letter and Author response
Decision letter https://doi.org/10.7554/eLife.29055.029
Author response https://doi.org/10.7554/eLife.29055.030

## Additional files

**Supplementary files**
• Transparent reporting form
DOI: https://doi.org/10.7554/eLife.29055.028

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
