## [Decision Letter]

Thank you for submitting your article "Striatal adenosine A_2A_ receptor neurons control active–period sleep via parvalbumin neurons in external globus pallidus" for consideration by *eLife*. Your article has been reviewed by three peer reviewers, one of whom is a member of our Board of Reviewing Editors and the evaluation has been overseen by Gary Westbrook as the Senior Editor. The reviewers have opted to remain anonymous. The reviewers have discussed the reviews with one another and the Reviewing Editor has drafted this decision to help you prepare a revised submission.

Summary:

Adenosine is an important molecule affecting sleep regulation. In this manuscript, the authors explore the roles of the striatal A_2A_R neurons in sleep/wake regulation. The authors found that animals tended to sleep more when A_2A_R neurons in different part of striatum (except the caudal part) were activated. This phenomenon fits with the known somnogenic effect of adenosine. In this study, the authors address two unexplored area of brain, and their interactive role in sleep–wake regulation. The study uses state of the art opto– and chemogenetic methodology to suggest that the A_2A_ receptor containing subpopulation of GABAergic neurons in rostral, central and lateral striatum project to the parvalbumin expressing neurons of globus pallidus. Activation of striatal population exerts an inhibitory tone onto GPe neurons and enhances NREM sleep in the dark (active) period only.

Essential revisions:

The authors declared that activation of A_2A_R neurons promoted NREM sleep through inhibition of PV neurons. The provided evidence could be stronger. In Figure 3, the author showed 'Optogenetic stimulation of the striatopallidal terminals inhibited GPe neurons' in brain slices. The authors might have tested whether opto–stimulation of the same terminals promotes NREM sleep in the freely moving animals. This issue requires further discussion.

Another critical piece of evidence is that lesion of PV neurons blocked the effects of A_2A_R neuron activation. However, activating of A_2A_R neurons (Figure 1) seemed to have a stronger effect than directly inhibiting PV neurons (Figure 4) suggesting PV neurons may not be the only downstream effectors. Sometimes ablation of neurons might not be a good option as it might cause a lot of stress in the local environment and therefore have a lot of potential side–effects. This might be the reason that ablation of PV neurons have the most dramatic effects on NREM sleep (Figure 5). A technical concern here is that the use of A_2A_R/PV double Cre mice. The author injected the virus at different locations and assumed the viruses will be expressed independently. This strategy is not clean because virus can diffuse from the injection sites (particularly in this scenario where striatum and GPe are relatively close to each other). In the meantime, A_2A_R neurons are shown to send projection to GPe. It is difficult to exclude the possibilities that a small portion of the A_2A_R neuron terminals might be infected by the Casp3 expressing virus which would confound the results. Again, these issues require careful discussion in the manuscript.

---

## [Author Response]

Essential revisions:The authors declared that activation of A_2A_R neurons promoted NREM sleep through inhibition of PV neurons. The provided evidence could be stronger. In Figure 3, the author showed 'Optogenetic stimulation of the striatopallidal terminals inhibited GPe neurons' in brain slices. The authors might have tested whether opto–stimulation of the same terminals promotes NREM sleep in the freely moving animals. This issue requires further discussion.

As requested, we tested the effects of photostimulation of ChR2–expressing A_2A_R neuron terminals in the GPe on sleep–wake behavior in freely moving mice. We found that 1–h optogenetic activation of ChR2–expressing A_2A_R neuron terminals with optical fiber implanted inside the GPe, significantly increased NREM sleep, compared with that during the same time period without stimulation. We added these new data into Figure 3—figure supplement 2–2D. Meanwhile, no significant change was observed in the EEG power density of NREM sleep during optogenetic stimulation (Figure 3—figure supplement 2). In the control mice only expressing mCherry in A_2A_R neurons, optogenetic stimulation of the striatopallidal terminals did not change the time spent in each stage and the EEG power density of NREM sleep (Figure 3—figure supplement 2–2J).

Another critical piece of evidence is that lesion of PV neurons blocked the effects of A_2A_R neuron activation. However, activating of A_2A_R neurons (Figure 1) seemed to have a stronger effect than directly inhibiting PV neurons (Figure 4) suggesting PV neurons may not be the only downstream effectors. Sometimes ablation of neurons might not be a good option as it might cause a lot of stress in the local environment and therefore have a lot of potential side–effects. This might be the reason that ablation of PV neurons have the most dramatic effects on NREM sleep (Figure 5). A technical concern here is that the use of A_2A_R/PV double Cre mice. The author injected the virus at different locations and assumed the viruses will be expressed independently. This strategy is not clean because virus can diffuse from the injection sites (particularly in this scenario where striatum and GPe are relatively close to each other). In the meantime, A_2A_R neurons are shown to send projection to GPe. It is difficult to exclude the possibilities that a small portion of the A_2A_R neuron terminals might be infected by the Casp3 expressing virus which would confound the results. Again, these issues require careful discussion in the manuscript.

Thank you for the comments. As suggested, we have noted the concern that activation of A_2A_R neurons (Figure 1) seemed to have a stronger effect than direct inhibition of GPe PV neurons (Figure 4). There are some possible explanations for the difference of increase in NREM sleep following the activation of A_2A_R neurons and inhibition of GPe PV neurons. Firstly, we compared the amount of NREM sleep during the –h post–injection period in *Adora2a*–Cre and *Pvalb*–Cre mice after vehicle and CNO injections (as shown in the Author response image 1). We found that *Pvalb*–Cre mice slept more than *Adora2a*–Cre mice after vehicle injection (P < 0.05). However, after CNO injection, the amount of NREM sleep increased to the same levels (P > 0.05). Since *Adora2a*–Cre and *Pvalb*–Cre mice have the different baseline or distinct responses to vehicle injections, we thought that the different lines of the transgenic mice may contribute to the different effect in NREM sleep. Secondly, we compared the amount of NREM sleep during the 3–h post–CNO injection period with that during the first 3–h (7 a.m.–10 a.m.) after light–on in *Adora2a*–Cre and *Pvalb*–Cre mice (as shown in the Author response image 1). We found that the amount of NREM sleep was almost at the same levels (P > 0.05) between the 3–h post–CNO injection period and the first 3–h after light–on during which mice often exhibit the highest amount of sleep in *Adora2a*–Cre and *Pvalb*–Cre mice. The present data suggested that CNO injection increased NREM sleep to reach the saturation of sleep during the inactive period after light–on. Lastly, chemogenetic activation of A_2A_R neurons induced the release of GABA as a neurotransmitter targeting at GABAA receptors in the GPe (Figure 3). Upon activation, GABAA receptors selectively conduct Cl^–^, resulting in hyperpolarization of GPe PV neurons (Johnston, 1996). However, CNO injection inhibited GPe PV neurons through the hM4Di receptors, which are mutant Gi coupled receptors. Upon activation, the hM4Di receptors subsequently activate G–protein–gated inwardly rectifying K^+^ channels, thus leading to hyperpolarization of the neuronal membrane (Dong et al., 2010). The difference between silencing GPe PV neurons through chemogenetic activation of striatal A_2A_R neurons and direct chemogenetic inhibition of GPe PV neurons may also contribute to the different effect on NREM sleep.

**Author response image 1. respfig1:** Total amounts of NREM sleep during the 3–h post–injection period (7 p.m.–10 p.m.) following vehicle (open columns) or CNO (blue columns) injections and during the first 3–h (7 a.m.–10 a.m.) after light–on (diagonal columns) in *Adora2a*–Cre mice expressing hM3Dq receptors in the rostral striatum and *Pvalb*–Cre mice expressing hM4Di receptors in the GPe. **P < 0.01; #P < 0.05; n.s.: no significance, *P* > 0.05.

We agree with the reviewers that sometimes ablation of neurons might not be a good option as it might cause a lot of stress in the local environment and therefore have a lot of potential side–effects. In the present study, we have noted that ablation of GPe PV neurons by using taCasp3 induced the apoptosis, and significantly increased NREM sleep in baseline conditions (revised Figure 5). This finding was in agreement with direct chemogenetic inhibition of GPe PV neurons (Figure 4), suggesting that PV neurons in the GPe are involved in wakefulness.

Using the *Adora2a*–Cre /*Pvalb*–Cre mice, we injected hSyn–DIO–hM3Dq–mCherry–AAV into the rostral and central striatum to express hM3Dq receptors (mCherry+) in A_2A_R neurons and Flex–taCasp3–TEVp–AAV into the GPe to induce the apoptosis in the PV neurons. Because the striatum and GPe are relatively close to each other, we could not absolutely exclude the possibility of diffusion of virus. However, in order to reduce the diffusion of virus between the striatum and GPe, the virus was microinjected into striatum and GPe at a very slow pace. After each experiment, the virus injection sites were confirmed. We observed that the hM3Dq–mCherry–AAV diffused into the caudal striatum since we detected a few mCherry positive cell bodies there, whereas we did not detect any mCherry positive cell body in the GPe except mCherry positive terminals, indicating that the GPe was not infected by virus expressing hM3Dq receptors. In addition, when the injected in the GPe, Flex–taCasp3–TEVp–AAV detected a small number of eGFP positive neurons in the caudal striatum, but not in the rostral and central striatum. Meanwhile, when the Flex–taCasp3–TEVp–AAV was injected in the GPe, it also diffused to the boundary between the caudal striatum and GPe, and expressed taCasp3 to induce the apoptosis of A_2A_R neurons in the small part of caudal striatum (Author response image 2). However, since A_2A_R neurons in the caudal striatum are not involved in sleep–wake regulation (Figure 1), we believed that the virus diffusion did not influence our results.

**Author response image 2. respfig2:** Micrographs of A_2A_R immunostaining in the rostral. (A, D), central (B, E), and caudal (C, F) striatum in the control group (A–C) injected eGFP–AAV and lesion group (D–F) injected taCasp3–AAV into the GPe of *Adora2a*/*Pvalb*–Cre mice. The black dotted line showed the boundary of the GPe, and the red dotted line showing the region of striatum infected by the taCasp3–AAV. Scale bar, 1 mm. Ctx, cortex; GPe, external globus pallidus.

The virus serotype of Flex–taCasp3–TEVp–AAV used in present study was serotype 9. As you indicated, several studies demonstrated that when injected in the striatum, entorhinal cortex, and dentate gyrus, the AAV 9 can infect the terminals projecting into the injection sites for retrograde transport (Castle, Gershenson, et al., 2014; Castle, Perlson, et al., 2014; Masamizu et al., 2011). In the present study, the mCherry–expression of the A_2A_R neuron terminals was robust in the GPe in the lesion group, as same as the control group (Figure 5), indicating that the apoptosis induced by Casp3 was limited in the GPe rather than striatal A_2A_R neurons. When control virus, the hSyn–DIO–eGFP–AAV9, was injected into the GPe, we did not detect eGFP expression in the rostral and central striatum, suggesting that the AAV 9 did not infect the terminals projecting from the striatum. Therefore, we believed that the Flex–taCasp3–TEVp–AAV hardly infected the terminals of the striatal A_2A_R neurons when the virus was injected in the GPe.